# AGENTSTUDIO: A TOOLKIT FOR BUILDING GENERAL VIRTUAL AGENTS

**Longtao Zheng**[1*]  **Zhiyuan Huang**[3*]  **Zhenghai Xue**[1]  **Xinrun Wang**[5]
**Bo An**[1,2]  **Shuicheng Yan**[2,4]
[1]Nanyang Technological University, Singapore  [2]Skywork AI  [3]ETH Zurich
[4]National University of Singapore  [5]Singapore Management University
https://ltzheng.github.io/agent-studio

## ABSTRACT

General virtual agents need to handle multimodal observations, master complex action spaces, and self-improve in dynamic, open-domain environments. However, existing environments are often domain-specific and require complex setups, which limits agent development and evaluation in real-world settings. As a result, current evaluations lack in-depth analyses that decompose fundamental agent capabilities. We introduce AgentStudio, a trinity of environments, tools, and benchmarks to address these issues. AgentStudio provides a lightweight, interactive environment with highly generic observation and action spaces, e.g., video observations and GUI/API actions. It integrates tools for creating online benchmark tasks, annotating GUI elements, and labeling actions in videos. Based on our environment and tools, we curate an online task suite that benchmarks both GUI interactions and function calling with efficient auto-evaluation. We also reorganize existing datasets and collect new ones using our tools to establish three datasets: GroundUI, IDMBench, and CriticBench. These datasets evaluate fundamental agent abilities, including GUI grounding, learning from videos, and success detection, pointing to the desiderata for robust, general, and open-ended virtual agents.

## 1 INTRODUCTION

Building general virtual agents that can utilize every software to enhance human productivity is a long-standing research challenge in artificial intelligence (Shi et al., 2017). Such agents perceive computer states as observations (e.g., screenshots) and take actions through human-computer interfaces (e.g., GUI, keyboard, and mouse) or function calling (e.g., APIs) in response to natural language instructions. Fueled by advancements in large language models (LLMs) (Brown et al., 2020; Chowdhery et al., 2022; OpenAI, 2023), there has been impressive progress in virtual agents (Zheng et al., 2023; Zhang et al., 2024; Tan et al., 2024). However, deploying these agents in real-world settings requires satisfying three key desiderata: (1) achieving **general grounding** by interacting with both GUIs and APIs and handling complex observations like videos, (2) possessing extensive **digital world knowledge** to complete tasks across diverse applications, and (3) self-improving and generalizing to unseen situations through trial and error in **open-ended** environments. For example, an agent exporting charts from a spreadsheet and adding them to a video needs to interact precisely with GUI elements in both the spreadsheet and the video editor. Using API actions can speed up the process, and video observations can help monitor and make adjustments. The agent needs knowledge of these applications, e.g., understanding their workflows, to effectively perform the task. If the video editor's layout is out of distribution, the agent should learn how to use it by interacting with the environment.

Therefore, the environments of virtual agents must (1) model various observation and action spaces to support all possible tasks, (2) provide online interactions for learning through trial and error, and (3) systematically benchmark agents' abilities to satisfy the three desiderata. However, many existing virtual agent benchmarks are static datasets (Deng et al., 2023; Rawles et al., 2023; Kapoor et al., 2024) or only consider domain-specific actions (e.g., web operations) (Zhou et al., 2023; Koh et al., 2024), which do not encompass the full range of tasks humans can perform. Recent environments

---

*Equal contribution.

| Desiderata for General Virtual Agents | AgentStudio Environments | AgentStudio Tools | AgentStudio Benchmarks | |
|---|---|---|---|---|
| General grounding | Video observations GUI & API actions | GUI element annotation | Online benchmark tasks | GroundUI |
| Digital world knowledge | Real-world devices | Video action annotation | | IDMBench |
| Open-ended learning | Interactive environments | Task creation & validation | | CriticBench |

Figure 1: AgentStudio focuses on three desiderata for general virtual agents. For general grounding in any software, it provides universal observation and action spaces, a GUI annotation tool, and the GroundUI dataset for enhancing and evaluating GUI grounding. To leverage digital knowledge, our tools enable recording human trajectories and annotating videos on real-world devices, which can be used to create datasets like IDMBench to evaluate agents' abilities to learn from videos. To support open-ended learning, AgentStudio offers interactive environments with language feedback, tools for task creation, and CriticBench to benchmark the ability for success detection. The overall agent performance is measured through an online task suite developed using our task creation tool.

have advanced toward more realistic settings (Xie et al., 2024; Rawles et al., 2024), but they still do not include video observations or allow taking actions through both GUIs and APIs. Moreover, they typically use success rates as the only metric, which may result in overoptimizing agent frameworks for specific tasks rather than real-world performance. Additionally, setting up these environments or customizing tasks within them is often non-trivial. As a result, current environments and benchmarks struggle to reliably evaluate agent performance in real-world scenarios. They are also difficult to expand to cover a broader range of tasks and provide detailed analyses of fundamental agent abilities.

AgentStudio provides an interactive, realistic, and lightweight environment with generic observation and action spaces, enabling agents to interact with arbitrary software (Figure 1). The observation space incorporates multiple modalities, ranging from screen recordings (videos) and screenshots (images) to code execution results (text). Agents can act through human-computer interfaces (e.g., keyboard-mouse operations) to control third-party applications, and perform function calling to interact with APIs. These features expand the task space to massively open-domain and real-world tasks typically performed by humans. The interactive nature of online environments allows agents to learn through trial and error, which is enhanced by the language feedback on failure reasons provided by our environment. We also integrate various tools for customizing and validating benchmark tasks, annotating GUI elements with bounding boxes, and labeling actions for actionless videos. Our implementation is designed to be lightweight, ensuring ease of installation, usage, and reproducibility.

Using AgentStudio environment and tools, we introduce an online task-completion benchmark and three datasets to evaluate fundamental agent abilities in real-world settings. The benchmark suite consists of 205 real-world tasks across various applications such as VS Code, Google Workspace, and Office suites. Solving these tasks requires agents to process complex observations, interact with GUIs, and perform function calling. We curate the benchmark through a three-stage process: (1) creating basic tasks and their auto-evaluators for single-application scenarios, (2) compositionally expanding tasks to cross-application workflows, and (3) manually validating task configurations and auto-evaluators. To improve evaluation efficiency, we implement over 100 functions for automatic environment resetting and evaluation. The three datasets, GroundUI, IDMBench, and CriticBench, focus on UI grounding, labeling actions in videos, and success detection, respectively. GroundUI includes 18K single-step instruction-screenshot pairs across various devices, with ambiguous or incorrect instructions recaptioned. Agents must generate precise UI element coordinates to complete these instructions. The other two datasets are based on multi-step agent trajectories. In IDMBench, models need to label actions in a video clip, reflecting their ability to leverage knowledge from video demonstrations. CriticBench evaluates models' ability to determine if a task has been successfully completed based on the trajectory, which is crucial for efficient evaluation and open-ended learning. Overall, these benchmarks are designed to be accessible and extensible, demonstrating the potential of AgentStudio for agent evaluation and in-depth analysis in real-world scenarios.

Experimental results indicate that even state-of-the-art vision-language models (VLMs) like GPT-4o experience significant performance declines in GUI and compositional tasks. For tasks that could be

completed using APIs, providing screenshot observations can lead to poorer performance compared to text-only observations, as models might be misled into using GUI actions instead of APIs. Notably, most existing models struggle with professional applications such as image editing software, whereas humans can solve 72.2% of those tasks. Evaluation results from the three datasets reveal that current general-purpose VLMs struggle to accurately predict the exact coordinates of GUI elements in screenshots. This inaccuracy accumulates over time, significantly reducing the success rate of multi-step tasks. We also find that as screen resolution increases, the accuracy decreases, leading to higher accuracy in mobile environments compared to web and desktop scenarios. Additionally, our findings suggest that current models are not yet effective in labeling actions in videos and struggle to reliably evaluate whether a trajectory is successful. Therefore, future work could enhance these abilities and refine agent frameworks to better utilize them.

## 2 AGENTSTUDIO ENVIRONMENT

Each computer task can be modeled as a partially observable Markov decision process (POMDP) $(\mathcal{S}, \mathcal{A}, \mathcal{O}, \mathcal{T}, \mathcal{R}, \mathcal{F}, \mathcal{U})$, with state space $\mathcal{S}$, action space $\mathcal{A}$, observation space $\mathcal{O}$, transition function $\mathcal{T}$, reward function $\mathcal{R}$, feedback space $\mathcal{F}$, and instruction space $\mathcal{U}$. AgentStudio offers an interactive and realistic environment with comprehensive observation space $\mathcal{O}$ and action space $\mathcal{A}$. The transition function $\mathcal{T}$ reflects the real-time dynamics of computer environments. The environment supports both scalar rewards from $\mathcal{R}$ and language feedback from $\mathcal{F}$ to facilitate open-ended LLM agents to self-correct (Shinn et al., 2023; Wang et al., 2023). Additionally, it comes with tools for creating benchmark tasks ($\mathcal{S}, \mathcal{A}, \mathcal{O}$, and $\mathcal{U}$), validating functional auto-evaluators ($\mathcal{R}$ and $\mathcal{F}$), annotating GUI elements, and labeling actions in videos. These features are designed according to the three desiderata for general virtual agents (Figure 1), aiming to promote research on agents that can handle real-time video observations, utilize both APIs and GUIs for improved efficiency, learn from video demonstrations, and adapt through environment interactions.

**Universal observation and action spaces.** AgentStudio features a comprehensive observation space $\mathcal{O}$ and action space $\mathcal{A}$ that mirror the complexity of real-world human-computer interactions. Specifically, the observation space is defined as a union of text, image, and video modalities, i.e., $\mathcal{O} = \mathcal{O}_{\text{Text}} \cup \mathcal{O}_{\text{Image}} \cup \mathcal{O}_{\text{Video}}$. This allows the agent to receive observations that include screen recording videos, screenshots, and code execution results. Agents can also leverage tools to obtain additional observations that are not directly visible to human users, such as HTML code and accessibility trees. The video observation facilitates research on agents' ability to process complex, multimodal observations and solve tasks requiring real-time video understanding. The action space is similarly expansive, defined as a union of GUI interactions and API calls, i.e., $\mathcal{A} = \mathcal{A}_{\text{GUI}} \cup \mathcal{A}_{\text{API}}$. This encompasses both high-level function calling ($\mathcal{A}_{\text{API}}$) and low-level operations like keyboard and mouse controls ($\mathcal{A}_{\text{GUI}}$). Agents can execute code via a Python kernel to call APIs, import tools, and manipulate human-computer interfaces. This universal action space enables agents to control arbitrary software: they can call APIs for applications with accessible internals (e.g., terminal commands and Google Workspace) to enhance efficiency and accuracy, or automate operations via human-computer interfaces when API access is unavailable (e.g., third-party applications). In contrast, most environments use domain-specific observation and action spaces, such as a screenshot ($\mathcal{O}_{\text{Image}}$) or HTML snippet ($\mathcal{O}_{\text{Text}}$) for each web operation ($\mathcal{A}_{\text{API}}$), as shown in Figure 2.

**Interactive, realistic, and lightweight environments with language feedback.** AgentStudio adopts fully online interactive environments on real-world devices, where both screen recording and action execution occur in *real time*. The transition function $\mathcal{T} : \mathcal{S} \times \mathcal{A} \rightarrow \mathcal{S} \times \mathcal{O}$ reflects these real-time changes, capturing the complexity of real-world scenarios. For example, when the agent clicks an icon, the observation may not display an immediate change, despite the application being in the process of launching. In contrast, existing environments sidestep this challenge by adding a delay when getting the observation after each action, which can miss dynamic interactions or delays in actual scenarios. Our interactive environment allows agents to explore, learn from environment interactions, and accumulate new skills over time. To enhance agent learning, the environment provides both a binary reward signal $\mathcal{R} : \mathcal{S} \rightarrow \{0, 1\}$ and language feedback $\mathcal{F} : \mathcal{S} \rightarrow \{\text{failure reason}, \text{success}\}$ that informs agents why a task may have failed. This promotes research on self-correction and open-ended learning for LLM agents. Online environments offer more effective benchmarks than static datasets, which can be exploited, rely on domain-specific action spaces, or cannot accept multiple valid solutions. The real-world feature also enables agent evaluation and data collection on any task that a

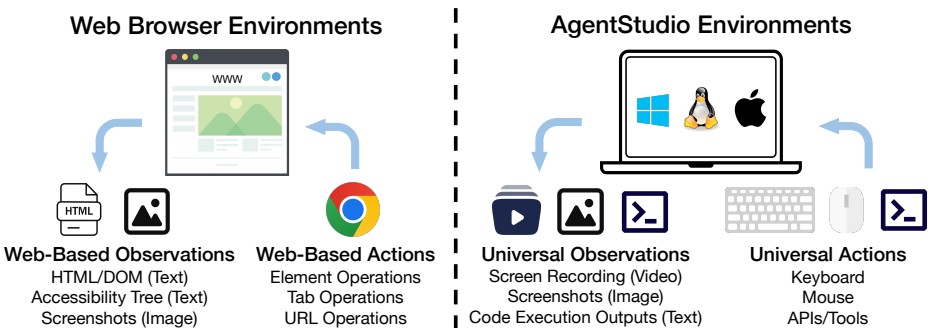

Figure 2: Comparison of observation and action spaces between web browser environments and AgentStudio. AgentStudio provides the most generic observation and action spaces, which significantly expands the task space, allowing for developing and evaluating agents in real-world settings.

human can perform. AgentStudio can operate in both local and remote modes. In remote mode, it launches a Virtual Network Computing (VNC) remote desktop, supporting various VNC-compatible operating systems (e.g., Windows, Linux, macOS) and devices (e.g., Docker containers, virtual machines, physical machines), making it lightweight and flexible to set up and use.

## 3   AGENTSTUDIO TOOLS

To facilitate the development and evaluation of agents within the AgentStudio environment, we provide three tools for creating in-the-wild benchmark tasks and developing high-quality datasets for agent training and evaluation. These tools, combined with the realistic environment of AgentStudio, contribute to the generation of rich, structured data for training and evaluating agents.

**Benchmark task creation and validation.** Users can create tasks by specifying $\mathcal{S}$, $\mathcal{A}$, $\mathcal{O}$, and $\mathcal{U}$, and equip these tasks with functional auto-evaluators representing $\mathcal{R}$ and $\mathcal{F}$. These tools enable manual validation of task performance and evaluator correctness, i.e., ensuring the accuracy of $\mathcal{R}$ and $\mathcal{F}$, as well as the measurement of human success rates and operation times.

**Step-level GUI element annotation.** This pipeline allows users to capture screenshots $o \in \mathcal{O}_{\text{Image}}$, annotate UI elements by drawing bounding boxes, and add step-level instructions $u \in \mathcal{U}$. This facilitates the creation of diverse datasets for training and evaluating single-step UI grounding models, which map instructions and screenshots to precise mouse click coordinates.

**Trajectory-level video-action recording and refinement.** This is a two-step process including action recording and video refining. During action recording, users can record the trajectory-level instruction $u \in \mathcal{U}$, the screen recording $o_{1:T} \in \mathcal{O}_{\text{Video}}$, and all keyboard and mouse operations $a_t \in \mathcal{A}_{\text{GUI}}$, where $t$ is a timestamp within a time horizon of length $T$. In the video refining phase, an editor interface allows users to replay and trim the recorded video $o_{1:T}$ and process the action sequence $a_{1:T-1}$. Specifically, users can remove noisy mouse movements and aggregate atomic actions (e.g., individual key presses) into higher-level actions (e.g., typing a string or executing a shortcut). This is crucial for training IDMs that can infer action sequences from videos without explicit action labels, enhancing the agent's ability of learning from video demonstrations.

## 4   AGENTSTUDIO OVERALL BENCHMARK TASKS

To illustrate how AgentStudio facilitates agent evaluation in complex and real-world scenarios, we introduce a benchmark suite consisting of 205 tasks. These tasks span API usages such as terminal and Gmail and GUI software like VS Code in the AgentStudio environment. Solving these tasks requires various fundamental agent abilities, including general grounding through complex action space ($\mathcal{A}_{\text{API}} \cup \mathcal{A}_{\text{GUI}}$) and compositional generalization over diverse instruction space $\mathcal{U}$. The tasks are designed to be easy to evaluate, circumventing common issues in current agent evaluations such as unrealistic environments, complex setups, and vulnerability to hacks. Despite their conceptual simplicity, these tasks present significant challenges to state-of-the-art VLMs.

## 4.1 BENCHMARK CONSTRUCTION

Each task in our benchmark can be formulated as a tuple $(u, s_0, \mathcal{R}, \mathcal{F})$, where $u \in \mathcal{U}$ is the natural language task instruction, $s_0 \in \mathcal{S}$ is the task-specific initial state, $\mathcal{R}$ automatically evaluates the final state and returns a binary reward, and $\mathcal{F}$ provides the language feedback if the task fails. For example, for the instruction $u$ "*Create a one-hour event Team Meeting in Google Calendar at 5pm today*", $s_0$ is obtained by removing any existing event with the same name, and $\mathcal{R}$ evaluates success by checking via API calls whether the event was correctly created after the agent's actions. If the task fails, $\mathcal{F}$ provides feedback on which success conditions are not satisfied for the final state. We source instructions from the Internet and previous work (Guo et al., 2023; Li et al., 2024; Xie et al., 2024) and construct the benchmark suite compositionally through the following three stages.

**Stage I: Crafting the minimal evaluators and action spaces.** We begin by organizing basic instructions into 50 single-API tasks and 95 single-GUI tasks, each focusing on API or GUI operations within a single application. We implement over 100 auto-evaluators for $\mathcal{R}$ and $\mathcal{F}$. Single-API tasks consist of tasks that can be accomplished through direct function calling, with the observation space $\mathcal{O} = \mathcal{O}_{\text{Text}}$. Single-GUI tasks involve common daily applications where agents are additionally provided with screenshots, i.e., $\mathcal{O} = \mathcal{O}_{\text{Text}} \cup \mathcal{O}_{\text{Image}}$. In both cases, we provide the agent with a minimal while generic action space $\mathcal{A} = \mathcal{A}_{\text{API}} \cup \mathcal{A}_{\text{GUI}}$, including basic keyboard, mouse, and terminal operations. We leave higher-level actions like skill libraries to future research on agent frameworks.

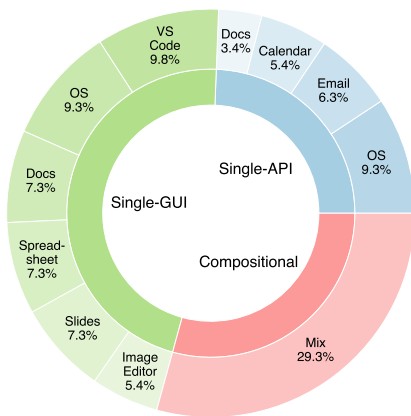

Figure 3: Task composition.

**Stage II: Compositional task expansion.** By compositionally extending the instructions and auto-evaluators from Stage I, we generate 60 more challenging tasks to benchmark compositional generalization. This category covers cross-application tasks requiring interactions with multiple GUIs, multiple APIs, or both. For example, agents may need to navigate the operating system and multiple applications, like VS Code and a document, to complete a task.

**Stage III: Manual validation and difficulty measure.** To verify the correctness of the auto-evaluators ($\mathcal{R}$ and $\mathcal{F}$), we use AgentStudio to collect human trajectories, evaluator outcomes, and task completion times. Therefore, we can identify and correct potential issues and mitigate false evaluations. The task completion times serve as empirical measures of task difficulty.

## 4.2 EVALUATION

| Model | Single-API | | Single-GUI | | | | Compositional | Total |
|---|---|---|---|---|---|---|---|---|
| | OS | Google | OS | Office | VS Code | GIMP | | |
| Claude 3.5 Sonnet | 94.7 | **74.2** | **94.7** | 0.0 | 5.0 | 0.0 | **25.0** | **36.6** |
| GPT-4o | **100.0** | 54.8 | **94.7** | **4.4** | **15.0** | 0.0 | 23.3 | 35.6 |
| Gemini 1.5 Pro | 68.4 | 16.1 | 63.2 | 0.0 | 5.0 | 0.0 | 5.0 | 16.6 |
| Gemini 1.5 Flash | 52.6 | 12.9 | 47.4 | 0.0 | 0.0 | 0.0 | 6.7 | 13.2 |
| Humans | 73.7 | 83.9 | 73.7 | 51.1 | 85.0 | 54.6 | 80.0 | 72.2 |

Table 1: Success rates (%) on AgentStudio overall benchmark tasks. Existing models can perform well in simple command-line tasks but struggle with complex API calls, GUI operations, and cross-application tasks that require both GUI operations and API calls.

Table 1 compares model performances on our benchmark tasks. While state-of-the-art models like GPT-4o perform well on simpler operating system tasks (Single-API OS) that only involve $\mathcal{O}_{\text{Text}}$, they struggle to reliably solve tasks that require complex API calls (e.g., Google Workspace). To evaluate the impact of the observation space, we apply identical instructions and action spaces in Single-API OS and Single-GUI OS, but Single-GUI OS tasks include additional screenshot observations, i.e., $\mathcal{O} = \mathcal{O}_{\text{Text}} \cup \mathcal{O}_{\text{Image}}$. Even though the tasks remain identical, the performance on the Single-GUI OS task declines for some models, as they are misled into using keyboard/mouse operations instead of

efficient API calls. Specifically, GPT-4o and Claude 3.5 Sonnet maintain comparable success rates across both setups, as they consistently favor API calls even when presented with screenshots. In contrast, Gemini models exhibit a performance drop, as they are prone to relying on GUI interactions. This emphasizes the limitations of current models and the need for agents to determine when to use APIs instead of GUI actions. Also, most existing models struggle with professional software applications, such as Office suites, code editor (VS Code), and image editor (GIMP). Furthermore, compositional tasks that span multiple applications or require a combination of GUI operations and API calls remain challenging for current models, despite being moderately difficult for humans. We manually review the trajectories and categorize five failure modes: (1) False Finish: The model incorrectly marked the task as complete but failed the auto-evaluation. (2) Parse Error: The model generated unparseable actions. (3) Step Limit: The model exceeded the allowed steps. (4) Time Limit: The model surpassed the time limit. (5) Repetition Limit: The model repeated the same action many times. Table 2 presents the distribution of failure modes for each model, and Table 3 shows the proportion of tasks actively marked as completed by each model. Gemini 1.5 Flash and Claude 3.5 Sonnet exhibit relatively low trigger rates for Step, Time, and Repetition Limits, indicating their capability to actively mark termination. In contrast, GPT-4o's active finish rate is significantly lower than the other models, suggesting that it tends to continue task execution until a certain limit or error is triggered. For most GUI tasks, existing models struggle to accurately click on GUI elements, as further demonstrated by our fine-grained evaluation in Section 5. When executing actions through code, existing models fail to handle dynamic real-time environments with appropriate sleep delays, and they struggle to determine when new observations are needed, leading to difficulties in completing complex tasks. Further results and detailed analysis are provided in Appendix F.

| Model | False Finish | Parse Error | Step Limit | Time Limit | Repetition Limit |
|---|---|---|---|---|---|
| Gemini 1.5 Pro | 70.8 | 2.9 | 11.7 | 9.4 | 5.3 |
| Gemini 1.5 Flash | 62.9 | 35.4 | 0.0 | 0.0 | 1.7 |
| Claude 3.5 Sonnet | 96.2 | 2.3 | 0.8 | 0.8 | 0.0 |
| GPT-4o | 31.8 | 48.5 | 4.5 | 8.3 | 6.8 |

| Model | Active Finish |
|---|---|
| Gemini 1.5 Pro | 74.1 |
| Gemini 1.5 Flash | 67.8 |
| Claude 3.5 Sonnet | 97.6 |
| GPT-4o | 28.8 |

Table 2: Distribution of failure modes for each model (%).          Table 3: Active finish rate (%).

## 5 AGENTSTUDIO FINE-GRAINED EVALUATION

To gain deeper insights into agent capabilities beyond the overall performance measured by online benchmark tasks, we develop three datasets using AgentStudio: GroundUI, IDMBench, and CriticBench. These datasets target the desiderata for general virtual agents, i.e., general UI grounding, learning from videos, and success detection. Unlike benchmark tasks where success rates are often zero, our datasets provide dense signals to guide research and mitigate overoptimizing specific tasks.

### 5.1 GROUNDUI: EVALUATING UI GROUNDING AND LOCALIZATION

Using AgentStudio, we confirm that accurately grounding UI elements with precise coordinates is still a main challenge for virtual agents (Li et al., 2020). As illustrated in Figure 4, we collect a sample task with over 200 screenshots from various desktop operating systems using AgentStudio. In this task, models are required to write code snippets that include click types (such as left click or double click) and the corresponding click locations. We find that current VLMs consistently perform poorly in matching locations accurately. Previous studies have also confirmed that while current models can generate high-level plans in textual form, they struggle to ground these plans into the correct UI elements (Zheng et al., 2024). This grounding capability

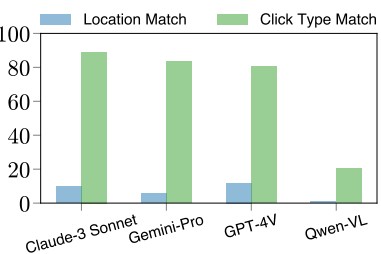

Figure 4: Accuracy of location/type match in the sample task.

is necessary because not all elements are readily accessible in text form (Koh et al., 2024). However, existing benchmarks have varying formats and do not directly evaluate this ability *across diverse applications and platforms*. More importantly, many instructions in these datasets are either incorrect

or ambiguous (see examples in Appendix C.2), and some instructions are task-level rather than step-level. These issues render existing datasets less reliable for benchmarking UI grounding. To address these issues, we systematically reorganize existing datasets and incorporated AgentStudio-collected data to create a more comprehensive and reliable evaluation.

**Dataset curation.** We define a UI grounding task as a tuple $(u, o, \text{bbox})$, where $u \in \mathcal{U}$ represents a single-step instruction, $o \in \mathcal{O}_{\text{Image}}$ is the screenshot, and $\text{bbox} = (x_1, y_1, x_2, y_2)$ denotes the bounding box of the targeted UI element. Given $u$ and $o$, the model is expected to output an action $a = (x, y)$, the normalized coordinates to click within the range $(0, 1)$. The action is considered correct if the predicted point lies within the bounding box. For example, if provided with a screenshot $o$ of a music player interface and the instruction $u$ is "*Click the Play button*", the predicted action $a$ should correspond to the bounding box of the Play button. We collect screenshots from the test sets of existing grounding datasets across web, desktop, and mobile platforms, including MoTIF (Burns et al., 2022),

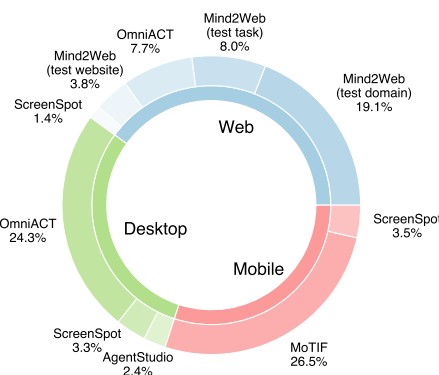

Figure 5: Source of GroundUI-1K data.

ScreenSpot (Cheng et al., 2024), OmniACT (Kapoor et al., 2024), along with additional screenshots collected in our sample task. After carefully examining the datasets, we exclude data lacking annotated bounding boxes. To recaption problematic instructions, we overlay the ground truth actions onto each screenshot and use GPT-4o to generate detailed descriptions for the plotted GUI elements, which proved to be reasonably reliable. This process results in GroundUI-18K, an 18K UI grounding dataset for benchmarking. For efficient evaluation, we conduct experiments on a subset, GroundUI-1K, which consists of 400, 300, and 300 samples for web, desktop, and mobile devices, respectively.

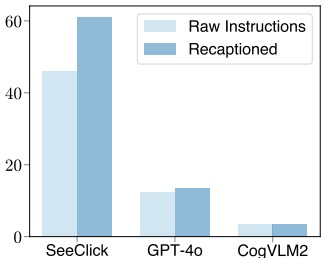

Figure 6: Accuracy w/ and w/o instruction recaptioning.

| Model | Web | Desktop | Mobile | Total |
|---|---|---|---|---|
| SeeClick | **64.3** | **44.3** | **73.7** | **61.1** |
| CogAgent | 25.3 | 15.7 | 35.7 | 25.5 |
| CogVLM2-Llama3-chat-19B | 2.5 | 2.7 | 5.3 | 3.4 |
| MiniCPM-Llama3-V 2.5 | 0.0 | 0.3 | 2.7 | 0.9 |
| Qwen-VL-Chat | 0.0 | 0.0 | 0.0 | 0.0 |
| Gemini 1.5 Pro | **31.2** | **24.3** | **51.3** | **35.2** |
| Claude 3.5 Sonnet | 13.0 | 14.0 | 26.3 | 17.3 |
| GPT-4o | 7.5 | 8.3 | 26.3 | 13.4 |
| Gemini 1.5 Flash | 0.5 | 4.3 | 26.3 | 9.4 |

Table 4: Single-step GroundUI-1K accuracy of open (top) and proprietary (bottom) VLMs.

**Results.** As presented in Table 4, SeeClick, a specialized UI grounding model, consistently achieves the highest overall grounding accuracy across different platforms. However, due to multi-step error accumulation, its performance still fails to meet the robustness required for practical deployment. SeeClick is trained with large-scale grounding data, highlighting the importance of scaling up UI grounding datasets. Among proprietary models, Gemini 1.5 Pro outperforms others like GPT-4o and Claude 3.5 Sonnet. However, these general-purpose models still lag behind specialized models such as SeeClick. The accuracy on mobile platforms is generally higher than that on web and desktop platforms, which can be attributed to the comparatively lower screen resolution of mobile screenshots. We also conduct ablation studies to assess the impact of instruction recaptioning on three models. As illustrated in Figure 6 in Appendix A, the differences in performance suggest that the recaptioned dataset provides a more reliable evaluation of UI grounding capabilities. Additional results, data samples, and prompts are available in Appendix C.

## 5.2 IDMBench: Evaluating Labeling Actions from Videos

Internet-scale data have significantly advanced language and vision models. However, virtual agents have not similarly benefited due to the lack of large-scale action datasets, because annotating computer operations is both time-consuming and expensive. While numerous tutorial videos exist

on the Internet, they lack the action labels necessary for training. Leveraging IDMs (Baker et al., 2022), which estimate $p_{IDM}(a_{1:T-1}|o_{1:T})$, allows us to label pseudo actions in videos, unlocking the knowledge embedded in Internet-scale videos. Unfortunately, no benchmark currently exists to evaluate this ability. Therefore, we evaluate VLMs as IDMs to promote research on enabling virtual agents to learn from videos and generalize across diverse scenarios.

**Dataset curation.** Our benchmark introduces two settings: IDM-Single and IDM-Multiple. IDM-Single focuses on single-step action labeling, where the task is to identify an action $a_t \in \mathcal{A}_{GUI}$ at timestep $t$ given the observations before and after it, denoted as $o_t$ and $o_{t+1}$. Although useful for benchmarking, selecting appropriate pairs of video frames in real-world scenarios is challenging because the effect of an action may not immediately appear after execution. Therefore, we propose IDM-Multiple to evaluate multi-step action labeling in a more realistic scenario. This setting requires the model to predict $p_{IDM}(a_{1:T-1}|o_{1:T})$, meaning it must label all actions performed within a trajectory of $T$ observations $o_t$ for $t \in [1, T]$, without being informed of the number of actions. We collect trajectories from Mind2Web (M2W) (Deng et al., 2023), Android in the Wild (AITW) (Rawles et al., 2023), VisualWebArena (VWA) (Koh et al., 2024), and AgentStudio (AS). Notably, we excluded AgentStudio-collected trajectories that involved code in actions due to the difficulty of evaluating their correctness, as well as trajectories with fewer than two steps. As a result, we created a dataset of 345 trajectories, with 100 from existing environments and 45 from AgentStudio. The IDM-Single dataset is derived from these trajectories by extracting individual transitions $(o, a, o')$. For simplicity, we formulate each task as a multi-choice question and incorporate the action space $A$ into the prompt as the answer choices. For example, in Mind2Web, the action space includes actions like clicking, typing, and scrolling. More details can be found in Appendix D.

| Model | IDM-Single | | | | | IDM-Multiple | | | | |
|---|---|---|---|---|---|---|---|---|---|---|
| | M2W | AITW | VWA | AS | Total | M2W | AITW | VWA | AS | Total |
| Claude 3.5 Sonnet | **73.0** | **56.0** | **50.0** | 72.0 | **61.4** | **18.0** | **8.0** | **7.0** | **22.2** | **12.5** |
| GPT-4o | 70.0 | **56.0** | 45.0 | **78.0** | 60.0 | 13.0 | **8.0** | 2.0 | 20.0 | 9.3 |
| Gemini 1.5 Pro | 62.0 | 51.0 | 46.0 | 48.0 | 52.3 | 0.0 | 0.0 | 1.0 | 2.2 | 0.6 |
| Gemini 1.5 Flash | 65.0 | 34.0 | 31.0 | 60.0 | 45.7 | 0.0 | 0.0 | 0.0 | 0.0 | 0.0 |
| Qwen-VL-Chat | 37.0 | 20.0 | 5.0 | 20.0 | 20.6 | 0.0 | 0.0 | 0.0 | 0.0 | 0.0 |

Table 5: Accuracy on IDMBench. Model performance drops significantly in IDM-Multiple.

**Results.** Table 5 presents the accuracy on IDMBench for various VLMs across four environments with different action spaces. Among the tested models, Claude 3.5 Sonnet and GPT-4o consistently outperform the others in both settings. Tasks involving more complex action spaces present greater challenges. For example, the action space of M2W is less complex than that of AITW, leading to higher accuracy. In IDM-Multiple, accuracy is calculated based on the exact match of the action sequences, which explains the notable decline in model performance, as any deviation from the expected sequence can lead to failure. Since models in IDM-Multiple can generate action sequences of variable lengths, Table 9 in Appendix A further evaluates the effort required to convert the predicted action sequence to the ground truth, using edit distance as a metric. Notably, two Gemini models sometimes fail to stop correctly and generate extremely long action sequences in IDM-Multiple. To mitigate this, we used a different prompt for IDM-Multiple experiments with these two models, instructing them to generate fewer than ten actions. However, Gemini 1.5 Flash still produced long sequences, resulting in the highest edit distance in IDM-Multiple. Overall, current models are still unable to accurately label actions from videos and are therefore not yet capable of learning from real-world tutorial videos. The discrepancy between performance in IDM-Single and IDM-Multiple also indicates that more effort is needed to improve action labeling in realistic settings.

## 5.3 CRITICBENCH: EVALUATING SUCCESS DETECTION

Although we have implemented auto-evaluators to assess functional correctness in our online benchmark, evaluating the success of all tasks in this manner is neither scalable nor feasible due to the massive diversity and open-ended nature of computer tasks. Therefore, a general critic model would significantly reduce the burden of creating online benchmark tasks. Such a model also enables agents to self-correct and engage in open-ended learning to master software when human demonstrations are limited. We argue that training a general critic is easier than training general agents. Unfortunately, no existing benchmarks measure this success detection ability in current VLMs for computer tasks.

**Dataset curation.** We represent each data point as a triplet $(u, \tau, y)$, where $u \in \mathcal{U}$ denotes the trajectory-level task instruction, $\tau$ is the corresponding trajectory consisting of multiple transitions $(o, a, o')$, and $y \in \{0, 1\}$ is the binary outcome indicating the success or failure of $\tau$. Given the instruction $u$ and the trajectory $\tau$, the model is required to predict $y$. Similar to IDMBench, we collect trajectories on different devices from M2W (Deng et al., 2023), AITW (Rawles et al., 2023), VWA (Koh et al., 2024), and AS. Since most human trajectories are successful, we balanced the dataset by generating failure cases from partial trajectories. For example, we can create a failure trajectory by extracting the first few steps of a successful one. We curate a subset of 350 trajectories, with 300 trajectories from existing environments and 50 from AgentStudio.

| Model | Observation-Action Pairs | | | | Observations Only | | | |
|---|---|---|---|---|---|---|---|---|
| | Web | Desktop | Mobile | Total | Web | Desktop | Mobile | Total |
| Gemini 1.5 Pro | **72.5** | 84.2 | 52.5 | **69.7** | 61.1 | **88.1** | 48.1 | 62.9 |
| Gemini 1.5 Flash | 65.6 | 85.2 | 57.8 | 67.3 | 64.1 | 75.0 | 52.5 | 63.1 |
| GPT-4o | 68.0 | 91.5 | 41.7 | 66.5 | 61.4 | 85.7 | 39.4 | 60.8 |
| Claude 3.5 Sonnet | 68.4 | **92.9** | 36.6 | 65.6 | **64.6** | 87.3 | 47.5 | 64.2 |
| Qwen-VL-Chat | 55.6 | 55.1 | **64.4** | 58.5 | 63.2 | 73.4 | **67.6** | **66.2** |

Table 6: F1 score on CriticBench. More metrics can be found in Table 10 in Appendix A.

**Results.** Table 6 presents the F1 scores for predicting the outcome $y$ using current VLMs that accept multi-image inputs. Additional metrics, including accuracy, precision, and recall, are provided in Table 10. These results indicate that existing models perform similarly on this task, with no significant discrepancy between open and proprietary models. Furthermore, unlike the benchmark results in GroundUI, where all models perform better on mobile environments, success detection on mobile environments does not appear to be easier than on other platforms. We also observe that incorporating action information offers some benefits, but the effects are limited, as illustrated in the right part of Table 6. In summary, while current VLMs show promise, their performance is still far from effective for self-improvement and open-ended learning. This benchmark provides a standard for measuring the success detection capabilities of current VLMs, facilitating further research in their prediction accuracy. Additional examples and prompts are provided in Appendix E.

## 6 RELATED WORK

| Environment | Domain | Obs. Space | Action Space | Inter-active | Data & Task Tools | Language Feedback | Light-weight | Decompose Abilities |
|---|---|---|---|---|---|---|---|---|
| World of Bits | Local Web | T/I/V | K/M | ✓ | ✗ | ✗ | ✓ | ✗ |
| WebShop | Local Web | T/I | Web Ops | ✓ | ✗ | ✗ | ✗ | ✗ |
| Mind2Web | Web Dataset | T/I | Web Ops | ✗ | ✗ | ✗ | ✓ | ✗ |
| WebArena | Local Web | T | Web Ops | ✓ | ✗ | ✗ | ✗ | ✗ |
| VisualWebArena | Local Web | T/I | Web Ops | ✓ | ✗ | ✗ | ✗ | ✗ |
| AndroidEnv | Mobile Emulator | T/I/V | Touchscreen | ✓ | ✗ | ✗ | ✗ | ✗ |
| AITW | Mobile Dataset | T/I | Touchscreen | ✗ | ✗ | ✗ | ✗ | ✗ |
| AndroidWorld | Mobile Emulator | T/I/V | Touchscreen | ✓ | ✗ | ✗ | ✗ | ✗ |
| OmniACT | Desktop Dataset | T/I | K/M | ✗ | ✗ | ✗ | ✓ | ✗ |
| OSWorld | Desktop | T/I | K/M | ✓ | ✗ | ✗ | ✗ | ✗ |
| **AgentStudio** | Desktop | T/I/V | K/M/Code | ✓ | ✓ | ✓ | ✓ | ✓ |

Table 7: AgentStudio supports online interactions with real-world operating systems using the most generic observation (text (T), image (I), and video (V)) and action spaces (keyboard (K), mouse (M), and code) among the listed environments. It offers benefits such as user-friendly GUI/video annotation and task customization, lightweight installation, and fine-grained datasets that decompose fundamental agent abilities.

**Environments for virtual agents.** Table 7 presents a comparison of our environment with other existing ones. Numerous efforts have been made to develop simulators for creating virtual agents that can automate computer tasks. For example, World of Bits (Shi et al., 2017) offered a minimalist web simulator, while AndroidEnv (Toyama et al., 2021) provided an emulator wrapper for Android devices. Early attempts in these environments primarily used deep reinforcement learning (Liu et al., 2018; Gur et al., 2018; Jia et al., 2018; Gur et al., 2021; Humphreys et al., 2022), which struggled to generalize beyond the environments. With novel capabilities like code as policies (Liang et al.,

2023) and tool use (Schick et al., 2024; Qin et al., 2023) of LLM-based virtual agents (Gur et al., 2023b; Kim et al., 2023; Zheng et al., 2023; Gur et al., 2023a), these simplified environments have been gradually solved. To build more realistic and complex settings, WebShop (Yao et al., 2022) introduced a web shopping environment. However, its tasks and observation/action spaces were browser-specific, i.e., $\mathcal{O} = \mathcal{O}_{\text{Text}} \cup \mathcal{O}_{\text{Image}}$ and $\mathcal{A} = \mathcal{A}_{\text{API}}$. Similar limitations apply to recent web environments like WebArena (Zhou et al., 2023) and VisualWebArena (Koh et al., 2024), where the action spaces are limited to web operations such as element clicking and URL navigation. Meanwhile, AndroidWorld (Rawles et al., 2024) focused specifically on Android devices. OSWorld (Xie et al., 2024) is a recent extension of WebArena to desktop environments focusing on GUI interactions, with $\mathcal{O} = \mathcal{O}_{\text{Text}} \cup \mathcal{O}_{\text{Image}}$ and $\mathcal{A} = \mathcal{A}_{\text{GUI}}$. In contrast, our environment supports video observations and both API and GUI actions, i.e., $\mathcal{O} = \mathcal{O}_{\text{Text}} \cup \mathcal{O}_{\text{Image}} \cup \mathcal{O}_{\text{Video}}$ and $\mathcal{A} = \mathcal{A}_{\text{GUI}} \cup \mathcal{A}_{\text{API}}$. This setting expands the task space and reveals new challenges for virtual agents, such as deciding when to use API or GUI actions to maximize efficiency and handling real-time dynamics. Additionally, we provide a more lightweight implementation, along with tools for benchmark task customization, open-domain GUI annotation, and video action annotation in real-world environments.

**Benchmarks for virtual agents.** Most environments above introduced accompanying benchmark tasks with functional auto-evaluation as the reward function $\mathcal{R}$. However, their task space is limited by the observation and action spaces of the environments. Also, most of them focused only on overall success rates, failing to promote research efforts towards fundamental agent capabilities. In contrast, our generic environment allows for any task humans can perform. While we source part of instructions from existing benchmarks, we add new tasks targeting the unique challenges posed by our environment, e.g., dealing with complex observation and action spaces. We also conduct rigorous checks on our auto-evaluators $\mathcal{R}$, fix errors in tasks within existing environments, and offer a simplified task structure and evaluation process. Additionally, we provide language feedback $\mathcal{F}$ on task failures for LLM agents. In addition to benchmark tasks in online environments, there are also static datasets designed to evaluate overall agent performance. AITW (Rawles et al., 2023) is a large-scale dataset focused on Android operations. Mind2Web (Deng et al., 2023) and WebLINX (Lù et al., 2024) evaluate web agents. Some benchmarks specifically measure agent performance in interacting with APIs (Xu et al., 2023; Liu et al., 2023; Mialon et al., 2023; Yao et al., 2024). More recent static benchmarks have also been developed for desktop environments (Cheng et al., 2024; Niu et al., 2024; Kapoor et al., 2024; Zhang et al., 2024; Wu et al., 2024). Unlike functional auto-evaluators in online environments, these datasets only accept a single successful trajectory, which limits their effectiveness in evaluating overall task completion, as done in existing benchmarks. We believe that static datasets are more appropriate for benchmarking specific agent abilities. While diverse datasets exist, a systematic approach is lacking to target fundamental agent abilities beyond measuring success rates. Although several GUI grounding-specific datasets similar to our GroundUI exist (Li et al., 2020; Burns et al., 2022), a unified format can significantly facilitate benchmarking across diverse applications and devices. As for IDMBench and CriticBench, no similar benchmarks currently exist. More detailed rationale of our datasets can be found in Appendix B.

## 7    DISCUSSION

**Conclusion.** AgentStudio provides a comprehensive solution for general virtual agents by offering interactive real-world environments, holistic tools, online benchmark tasks, and datasets for specific agent capabilities. Our design choices follow the desiderata for such agents: general grounding to interact with any software, digital world knowledge from Internet-scale video tutorials, and open-ended learning for self-improvement and generalization to new situations. We hope this work offers practical insights and promotes the evaluation and development of next-generation virtual agents.

**Limitations and future work.** We acknowledge certain limitations in our work. First, despite rigorous examinations, there might still be cases where auto-evaluators make incorrect judgments due to real-world complexity, a common issue of existing real-world benchmarks. Developing a general critic performing well on our CriticBench could alleviate this issue. Second, the GroundUI datasets might still have problematic instructions due to the automatic recaptioning process using GPT-4o. Third, our datasets are designed for easy evaluation but might be vulnerable to hacks. Continually updating them can mitigate issues like data leakage. Finally, although our implementation is compatible with real-world devices, our results for online benchmark tasks are obtained within an Ubuntu Docker environment, a tradeoff between reproducibility and real-world complexity.

REPRODUCIBILITY STATEMENT

All resources, such as code and datasets, are publicly available at our project page. The benchmark tasks and datasets are also hosted on Hugging Face.

ETHIC STATEMENT

Autonomous agents that can control real-world digital devices have inherent risks, such as deleting important files and sending spam emails. To mitigate these risks, our framework allows users to examine and confirm each action before execution. Users should take extra caution when using these agents to interact with real-world environments.

ACKNOWLEDGMENTS

This research is supported by the National Research Foundation Singapore and DSO National Laboratories under the AI Singapore Programme (AISGAward No: AISG2-GC-2023-009), NUS Start-up Grant A-0010106-00-00, and Singapore Ministry of Education (MOE) Academic Research Fund (AcRF) Tier 1 grant (No. MSS24C005).

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

## A   ADDITIONAL RESULTS

We provide complete results of GroundUI-1K (Table 8), the edit distance of IDM-Multiple (Table 9), more metrics in CriticBench (Table 10), and application-wise results of the UI grounding sample task collected with AgentStudio (Table 11). Figure 7 analyzes the relationship between element size and accuracy. It suggests that larger element size may lead to higher grounding accuracy and explains why current models tend to perform better in mobile settings compared to web and desktop scenarios. In all experiments, we utilize the 001 version of the Gemini models and the 20240620 version of Claude 3.5 Sonnet. For online benchmark tasks, we use GPT-4o (0806), while for other results, we use GPT-4o (0513). The results presented in the paper were obtained using greedy decoding, with the temperature set to 0. For non-GUI tasks, we limit the maximum number of execution steps to 1. For GUI tasks, we limit the maximum number of execution steps to 30. For non-GUI tasks, we limit the maximum execution time to 30 seconds. For GUI tasks, we limit the maximum execution time to 60 seconds. If the model performs repeated actions, i.e., executes the same action consecutively three times, the task is considered a failure.

| Model | Web | Desktop | Mobile | Total |
|---|---|---|---|---|
| **Open Models** | | | | |
| SeeClick | **64.3** | **44.3** | **73.7** | **61.1** |
| CogAgent | 25.3 | 15.7 | 35.7 | 25.5 |
| CogVLM2-Llama3-chat-19B | 2.5 | 2.7 | 5.3 | 3.4 |
| MiniCPM-Llama3-V 2.5 | 0.0 | 0.3 | 2.7 | 0.9 |
| Qwen-VL-Chat | 0.0 | 0.0 | 0.0 | 0.0 |
| PaliGemma-3B-mix-448 | 0.0 | 0.0 | 0.0 | 0.0 |
| PaliGemma-3B-896 | 0.0 | 0.0 | 0.0 | 0.0 |
| **Proprietary Models** | | | | |
| Gemini 1.5 Pro | **31.2** | **24.3** | **51.3** | **35.2** |
| Claude 3.5 Sonnet (0620) | 13.0 | 14.0 | 26.3 | 17.3 |
| GPT-4o (0513) | 7.5 | 8.3 | 26.3 | 13.4 |
| GPT-4 Turbo (0409) | 5.3 | 11.0 | 23.0 | 12.3 |
| Gemini 1.5 Flash | 0.5 | 4.3 | 26.3 | 9.4 |
| Gemini 1.0 Pro | 0.5 | 0.3 | 5.0 | 1.8 |

Table 8: Complete accuracy on GroundUI-1K.

| Model | Mind2Web | AITW | VWA | AgentStudio | Total |
|---|---|---|---|---|---|
| Claude 3.5 Sonnet | **2.0** | **2.1** | **2.9** | **1.6** | **2.3** |
| GPT-4o (0513) | 2.1 | 2.2 | 3.5 | 2.0 | 2.5 |
| Gemini 1.5 Pro | 6.0 | 4.4 | 7.0 | 3.8 | 5.5 |
| Qwen-VL-Chat | 5.1 | 15.4 | 5.8 | 6.3 | 8.4 |
| Gemini 1.5 Flash | 294.5 | 7.2 | 7.2 | 7.8 | 90.6 |

Table 9: Edit Distance on IDM-Multiple (↓). The edit distance is the number of operations (insertion, deletion, and replacement) needed to convert the predicted action sequences to the ground truth.

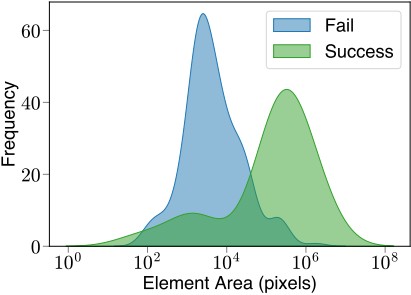

Figure 7: The distribution of element areas between successful and failed clicks in AgentStudio sample task. Tasks with smaller elements tend to fail.

| Model | Observation-Action Pairs | | | | Only Observations | | | |
|---|---|---|---|---|---|---|---|---|
| | Web | Desktop | Mobile | Total | Web | Desktop | Mobile | Total |
| **Accuracy** | | | | | | | | |
| Gemini 1.5 Pro | **73.5** | 82.0 | 62.0 | **71.4** | 65.0 | **86.0** | 59.0 | 66.3 |
| Gemini 1.5 Flash | 68.5 | 82.0 | **65.0** | 69.4 | **67.0** | 72.0 | **62.0** | 66.3 |
| Claude 3.5 Sonnet | 70.0 | **92.0** | 55.0 | 68.9 | 66.0 | **86.0** | 58.0 | **66.6** |
| GPT-4o (0513) | 68.5 | 90.0 | 58.0 | 68.6 | 63.5 | 84.0 | 57.0 | 64.6 |
| Qwen-VL-Chat | 52.0 | 38.0 | 48.0 | 48.9 | 54.5 | 58.0 | 52.0 | 54.3 |
| **F1 Score** | | | | | | | | |
| Gemini 1.5 Pro | **72.5** | 84.2 | 52.5 | **69.7** | 61.1 | **88.1** | 48.1 | 62.9 |
| Gemini 1.5 Flash | 65.6 | 85.2 | 57.8 | 67.3 | 64.1 | 75.0 | 52.5 | 63.1 |
| GPT-4o (0513) | 68.0 | 91.5 | 41.7 | 66.5 | 61.4 | 85.7 | 39.4 | 60.8 |
| Claude 3.5 Sonnet | 68.4 | **92.9** | 36.6 | 65.6 | **64.6** | 87.3 | 47.5 | 64.2 |
| Qwen-VL-Chat | 55.6 | 55.1 | **64.4** | 58.5 | 63.2 | 73.4 | **67.6** | **66.2** |
| **Precision** | | | | | | | | |
| Gemini 1.5 Pro | **75.3** | 88.9 | 70.0 | **76.7** | 68.8 | 89.7 | 65.5 | **72.5** |
| Claude 3.5 Sonnet | 72.2 | **100.0** | 61.9 | 75.9 | 67.4 | **96.0** | 63.3 | 71.4 |
| Gemini 1.5 Flash | 72.3 | 83.9 | **72.7** | 74.8 | **70.2** | 80.8 | **70.0** | 72.1 |
| GPT-4o (0513) | 69.1 | 93.1 | 68.2 | 73.6 | 65.2 | 92.3 | 66.7 | 70.6 |
| Qwen-VL-Chat | 51.7 | 48.7 | 49.0 | 50.2 | 53.1 | 59.2 | 51.0 | 53.4 |
| **Recall** | | | | | | | | |
| Qwen-VL-Chat | 60.0 | 63.3 | **94.0** | 70.0 | **78.0** | 96.7 | **100.0** | **87.2** |
| Gemini 1.5 Pro | **70.0** | 80.0 | 42.0 | 63.9 | 55.0 | 86.7 | 38.0 | 55.6 |
| Gemini 1.5 Flash | 60.0 | 86.7 | 48.0 | 61.1 | 59.0 | 70.0 | 42.0 | 56.1 |
| GPT-4o (0513) | 67.0 | **90.0** | 30.0 | 60.6 | 58.0 | 80.0 | 28.0 | 53.3 |
| Claude 3.5 Sonnet | 65.0 | 86.7 | 26.0 | 57.8 | 62.0 | 80.0 | 38.0 | 58.3 |

Table 10: Accuracy, F1 score, precision, and recall on CriticBench.

| Model | Windows | | | | Linux | | | MacOS | |
|---|---|---|---|---|---|---|---|---|---|
| | OS | Games | PowerPoint | Word | OS | Browser | Calculator | Music | iMovie |
| SeeClick | 42.3 | 47.6 | 24.0 | 9.1 | 60.7 | 9.1 | 69.0 | 53.8 | 35.7 |
| Claude-3 Sonnet | 3.8 | 4.8 | 24.0 | 9.1 | 7.1 | 4.5 | 0.0 | 30.8 | 0.0 |
| GPT-4V (1106) | 11.5 | 23.8 | 20.0 | 9.1 | 14.3 | 4.5 | 3.4 | 11.5 | 3.6 |
| Gemini-1.0 Pro | 7.7 | 9.5 | 8.0 | 4.5 | 10.7 | 0.0 | 0.0 | 3.8 | 0.0 |
| Qwen-VL-Chat | 0.0 | 0.0 | 0.0 | 0.0 | 0.0 | 0.0 | 0.0 | 0.0 | 0.0 |

Table 11: Application-wise results on AgentStudio-collected GroundUI sample task.

# B DESIDERATA FOR GENERAL VIRTUAL AGENTS

In this section, we elaborate on the three essential pillars for building next-generation virtual agents that can generalize and self-improve. The rationale behind AgentStudio is to provide benchmarks covering the desiderata for such agents, as well as useful environments and tools to pave the way for the emergence of general virtual agents. Specifically, we need online environments with generic observation/action spaces and real-world interactions (Section 2), tools that facilitate agent benchmarking and data annotation (Section 3), and benchmarks that both measure overall agent performance (Section 4) and decompose these desiderata (Section 5).

**General UI grounding in the wild.** Though UI grounding is a long-standing research problem and is considered a relatively simple task, it is still one of the major issues of current virtual agents. It has been validated that an oracle grounding can drastically improve agent performance (Zheng et al., 2024). However, most existing benchmarks focused on the grounding tasks of mobile phones (Burns et al., 2022; Rawles et al., 2023). Though there are some recent datasets of diverse trajectories in web and desktop environments (Deng et al., 2023; Lù et al., 2024; Kapoor et al., 2024), they have different formats and accessibility, which are not readily applicable for evaluating the grounding abilities of new models. For example, a recording in these datasets often consists of a high-level instruction and the corresponding human trajectory. However, for the grounding action at each step, the UI element may not directly relate to the instruction. Also, we note that there are a lot of ambiguous or incorrect instructions within those grounding-specific datasets, which can result in inaccurate evaluation. Therefore, to facilitate research on UI grounding in the wild, it is necessary to curate a

diverse and cross-platform benchmark with clear instructions, as well as a scalable data pipeline for both finetuning and benchmarking purposes. Additionally, many current online environments often overlook incorporating videos in the agent observation modality, where agents can fail to capture real-world dynamic changes and solve tasks that require monitoring the screen.

**Learning from Internet-scale actionless videos.** Internet-scale data have contributed to remarkable success in language and vision. The development of virtual agents can also benefit from the wealth of instruction videos available on the web, similar to MineCLIP (Fan et al., 2022) and VPT (Baker et al., 2022). Compared to recording human trajectories, collecting videos is also cheaper and more scalable. However, one of the main challenges for current agents to benefit from video data is the lack of action labels in videos. Unfortunately, there is currently no relevant computer agent benchmark and infrastructure to guide research in learning to act from videos (i.e., inverse dynamics models). Such models can add pseudo action labels to video recordings to unlock the knowledge within Internet-scale videos, enabling virtual agents to generalize.

**Open-ended learning with universal critic.** It is not scalable and feasible to examine whether a task is successful through task-specific, hand-written rules, because computer tasks are massively diverse and open-domain. Though it is challenging for current models to master these tasks, it may be easier for models to do success detection first (i.e., discriminating trajectory outcomes as a critic model). A general critic model allows agents to bootstrap via reinforcement learning or tree search. It can also facilitate agent self-correction if the model can provide natural language feedback. Compared to binary scalar rewards widely used in previous web environments and reinforcement learning environments, natural language feedback is considered more helpful for language agents to fix their own mistakes (Shinn et al., 2023). This ability is particularly significant for software where human demonstrations are limited, but there exist no benchmarks measuring it.

## C    DETAILS OF GROUNDUI

### C.1    DATA SAMPLE

Figure 8 is an example of the data in GoundUI datasets. We deliberately curated a host of existing datasets for virtual agents across different platforms from diverse datasets. ScreenSpot (Cheng et al., 2024) is a dataset with 610 screenshots. Based on these screenshots, 1,272 instructions are collected, with 502/334/436 for mobile/desktop/web environments, respectively. In addition to ScreenSpot, we collect 9,268 single-step data from Mind2Web test sets, 3804 from OmniACT test sets (Kapoor et al., 2024), 3,455 from MoTIF test sets (Burns et al., 2022), 1,272 from ScreenSpot benchmark, and 227 annotated with AgentStudio tools. There are 18,026 data entries in total, comprising 13,522 unique screenshots. After deliberately examining existing datasets, the data without UI elements annotated in bounding boxes are filtered out.

### C.2    RECAPTIONING AMBIGUOUS AND INCORRECT INSTRUCTIONS

Many existing UI grounding datasets consist of language instructions automatically inferred from non-readable internal meta information and did not go through rigorous cleaning. Some of the instructions are ambiguous, incorrect, or N/A, and not human-understandable. This makes the evaluation results of agent grounding on these datasets inaccurate. Therefore, we recaption the instructions with the help of GPT-4o. Figure 9, Figure 10, Figure 11, and Figure 12 demonstrate the examples of recaptioning in mobile, desktop, and web environments. The recaptioning process on GroundUI-1K costs 1M input tokens and 9K output tokens in total.

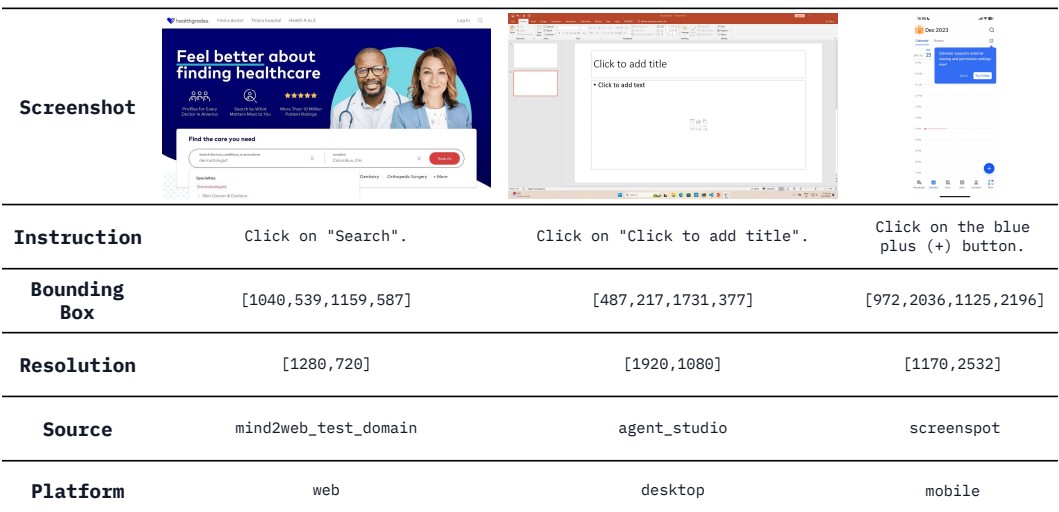

| | | | |
|---|---|---|---|
| **Screenshot** | | | |
| **Instruction** | Click on "Search". | Click on "Click to add title". | Click on the blue plus (+) button. |
| **Bounding Box** | [1040,539,1159,587] | [487,217,1731,377] | [972,2036,1125,2196] |
| **Resolution** | [1280,720] | [1920,1080] | [1170,2532] |
| **Source** | mind2web_test_domain | agent_studio | screenspot |
| **Platform** | web | desktop | mobile |

Figure 8: Data samples in GroundUI-1K/18K.

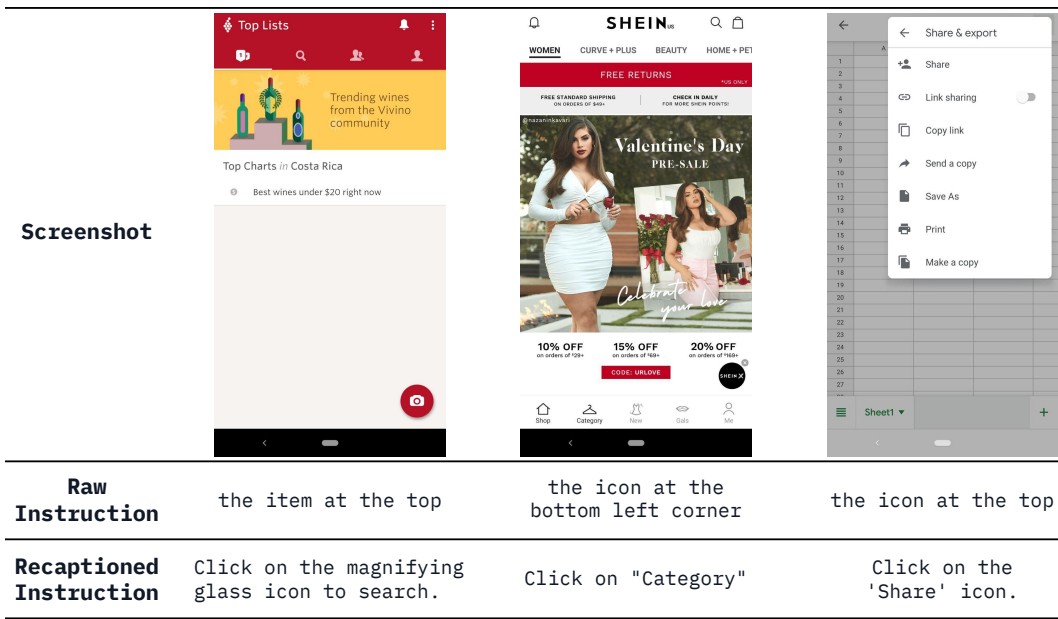

| | | | |
|---|---|---|---|
| **Screenshot** | | | |
| **Raw Instruction** | the item at the top | the icon at the bottom left corner | the icon at the top |
| **Recaptioned Instruction** | Click on the magnifying glass icon to search. | Click on "Category" | Click on the 'Share' icon. |

Figure 9: Examples of ambiguous instructions on mobile platforms sampled from MoTIF (Burns et al., 2022). The recaptioned instructions on GroundUI-1K are shown under the raw instructions from the original dataset.

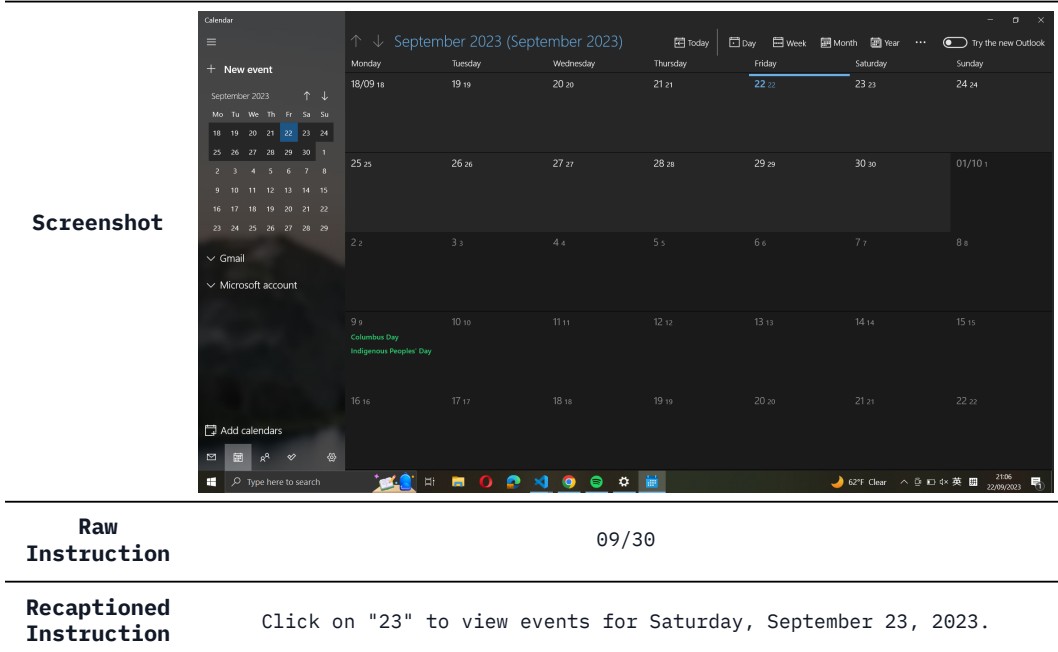

| | |
|---|---|
| **Screenshot** | (calendar screenshot) |
| **Raw Instruction** | 09/30 |
| **Recaptioned Instruction** | Click on "23" to view events for Saturday, September 23, 2023. |

Figure 10: Example of incorrect instructions on desktop platforms sampled from OmniACT (Kapoor et al., 2024). The recaptioned instructions on GroundUI-1K are shown under the raw instructions from the original dataset.

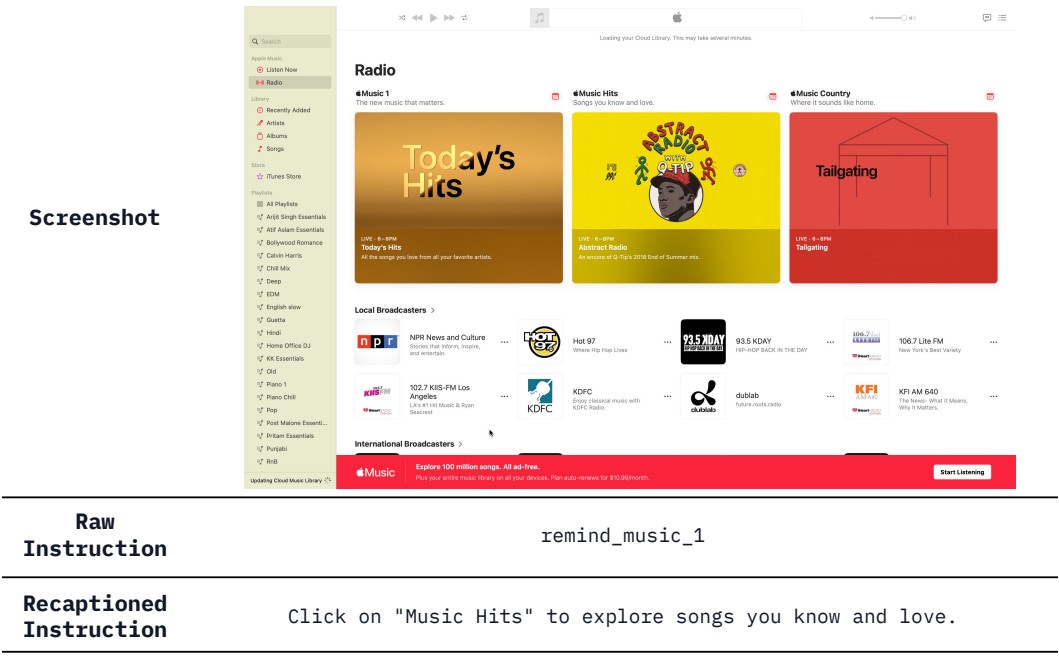

| | |
|---|---|
| **Screenshot** | (music radio screenshot) |
| **Raw Instruction** | remind_music_1 |
| **Recaptioned Instruction** | Click on "Music Hits" to explore songs you know and love. |

Figure 11: Example of ambiguous instructions on desktop platforms sampled from OmniACT (Kapoor et al., 2024). The recaptioned instructions on GroundUI-1K are shown under the raw instructions from the original dataset.

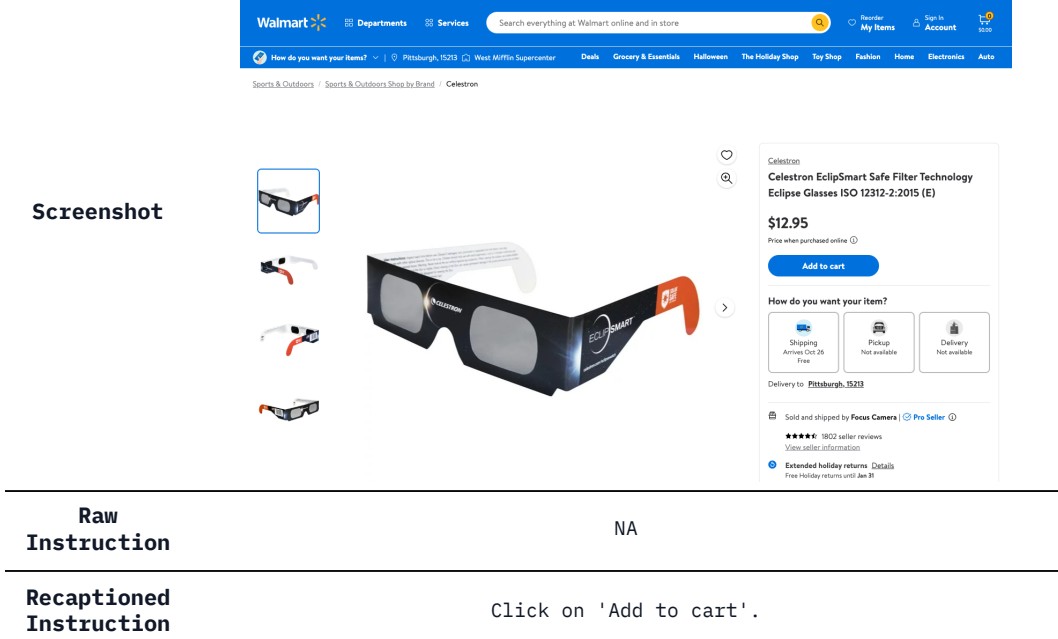

| | |
|---|---|
| **Screenshot** | |
| **Raw Instruction** | NA |
| **Recaptioned Instruction** | Click on 'Add to cart'. |

Figure 12: Example of incorrect instructions on web sampled from OmniACT (Deng et al., 2023). The recaptioned instructions on GroundUI-1K are shown under the raw instructions from the original dataset.

## C.3 Prompts for GroundUI

The prompts for GroundUI experiments are shown below. We match the model output with a regular expression. We use the response from Claude 3.5 Sonnet for demonstration.

**User**

`<datasets/gui_grounding/images/web_amc_screen1.png>`

Please output the coordinate for the next action based on the instruction and screenshot. Your answer should be of the following format: '(X, Y)' (without quotes) where X, Y is the coordinates ranging from 0 to 1.
Instruction: Click on "Discounts".
Answer:

**Assistant**
Based on the screenshot and the instruction to click on "Discounts", the approximate coordinates would be:
(0.85, 0.07)
This location corresponds to where the "Discounts" link appears in the top menu bar of the AMC website, slightly right of center near the top of the image.

**Evaluation**
AgentStudio will parse the response into (0.85, 0.07). This coordinate will be multiplied by the screenshot resolution to (1224.0, 63.0), which is outside the ground truth bounding box (1064, 99, 1148, 140) (left, top, right, bottom). Therefore, this result is incorrect.

# D DETAILS OF IDMBENCH

## D.1 DATA SAMPLE

Figure 13, Figure 14, and Figure 15 are three examples of IDM-Single dataset on different platforms. Figure 16 and Figure 17 are two examples for IDM-Multiple dataset.

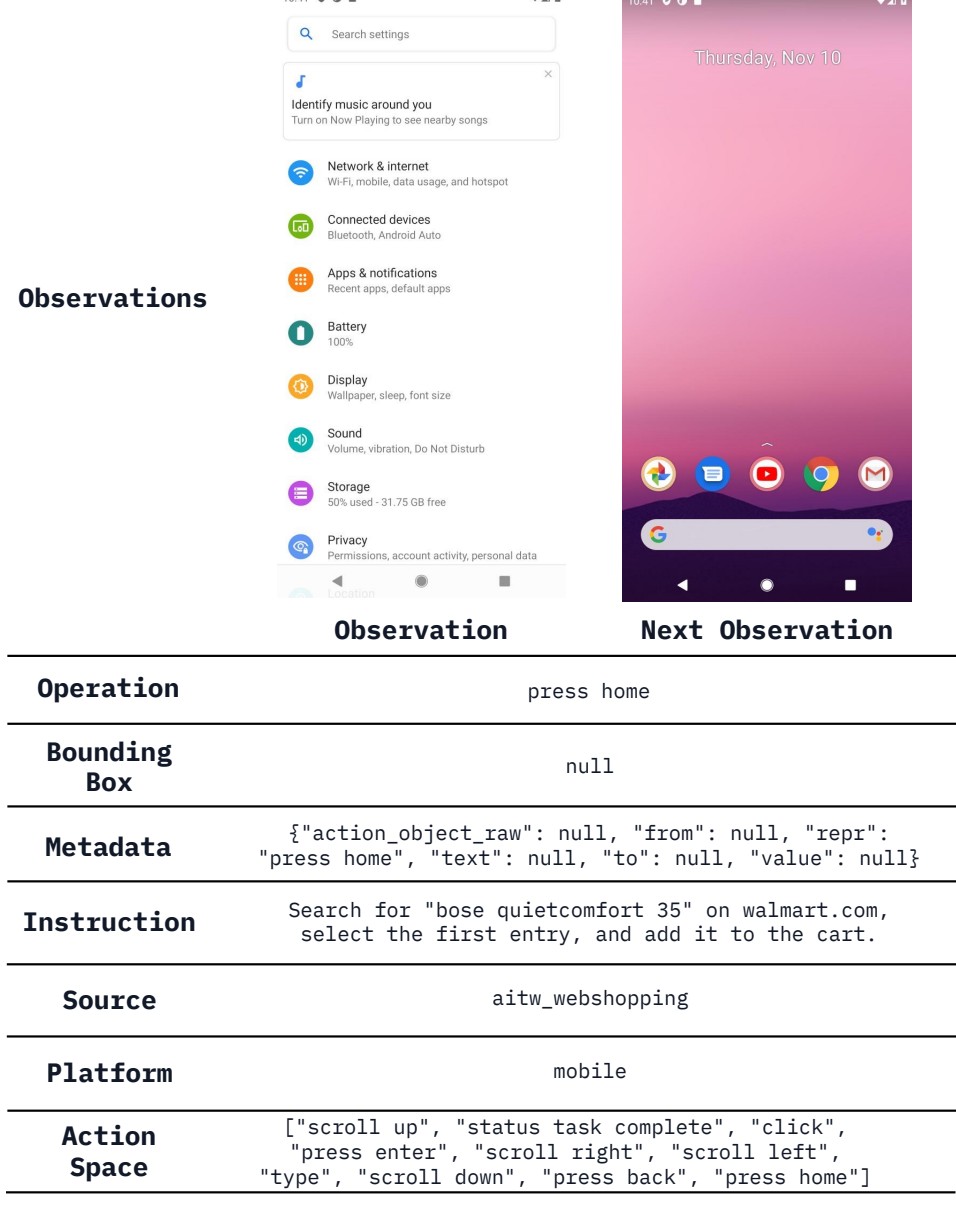

| | | |
|---|---|---|
| **Operation** | press home | |
| **Bounding Box** | null | |
| **Metadata** | {"action_object_raw": null, "from": null, "repr": "press home", "text": null, "to": null, "value": null} | |
| **Instruction** | Search for "bose quietcomfort 35" on walmart.com, select the first entry, and add it to the cart. | |
| **Source** | aitw_webshopping | |
| **Platform** | mobile | |
| **Action Space** | ["scroll up", "status task complete", "click", "press enter", "scroll right", "scroll left", "type", "scroll down", "press back", "press home"] | |

Figure 13: Data sample of pressing home button on mobile platform in IDM-Single dataset.

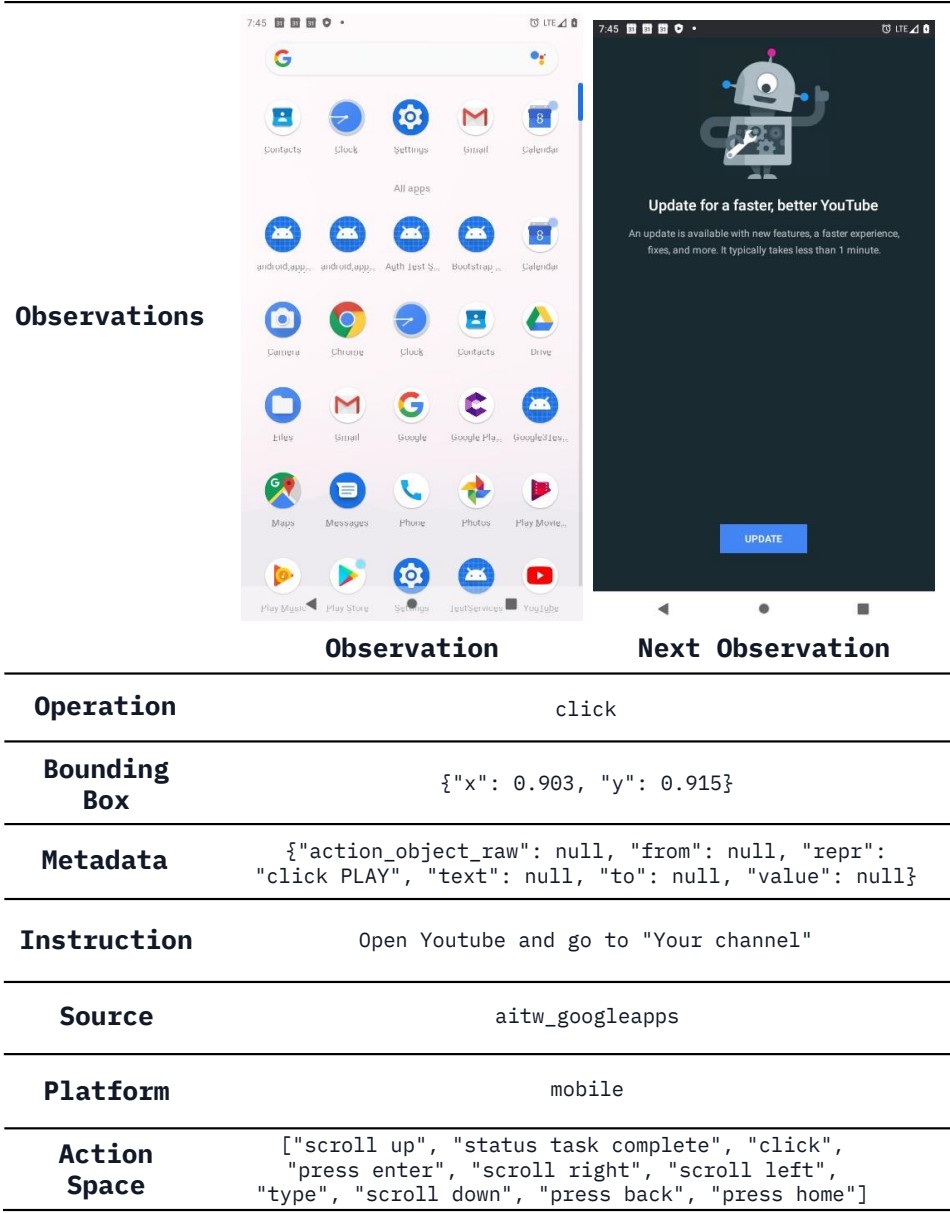

Figure 14: Data sample of touching (clicking) on mobile platform in IDM-Single dataset.

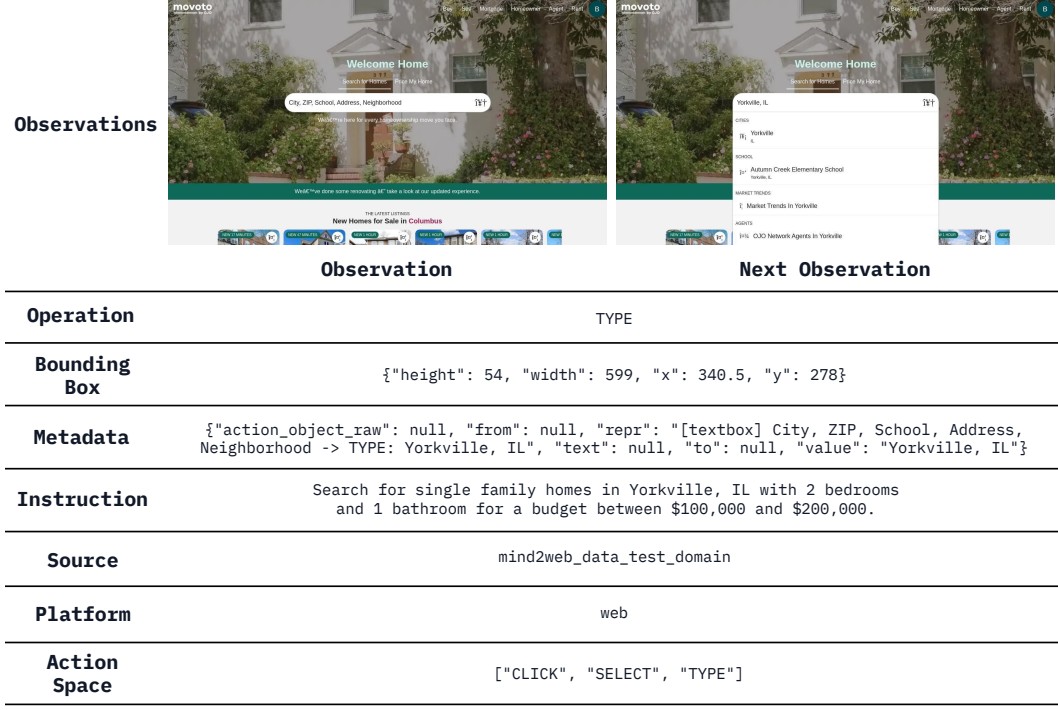

| Observations | Observation | Next Observation |
|---|---|---|
| Operation | TYPE | |
| Bounding Box | {"height": 54, "width": 599, "x": 340.5, "y": 278} | |
| Metadata | {"action_object_raw": null, "from": null, "repr": "[textbox] City, ZIP, School, Address, Neighborhood -> TYPE: Yorkville, IL", "text": null, "to": null, "value": "Yorkville, IL"} | |
| Instruction | Search for single family homes in Yorkville, IL with 2 bedrooms and 1 bathroom for a budget between $100,000 and $200,000. | |
| Source | mind2web_data_test_domain | |
| Platform | web | |
| Action Space | ["CLICK", "SELECT", "TYPE"] | |

Figure 15: Data sample on web in IDM-Single dataset.

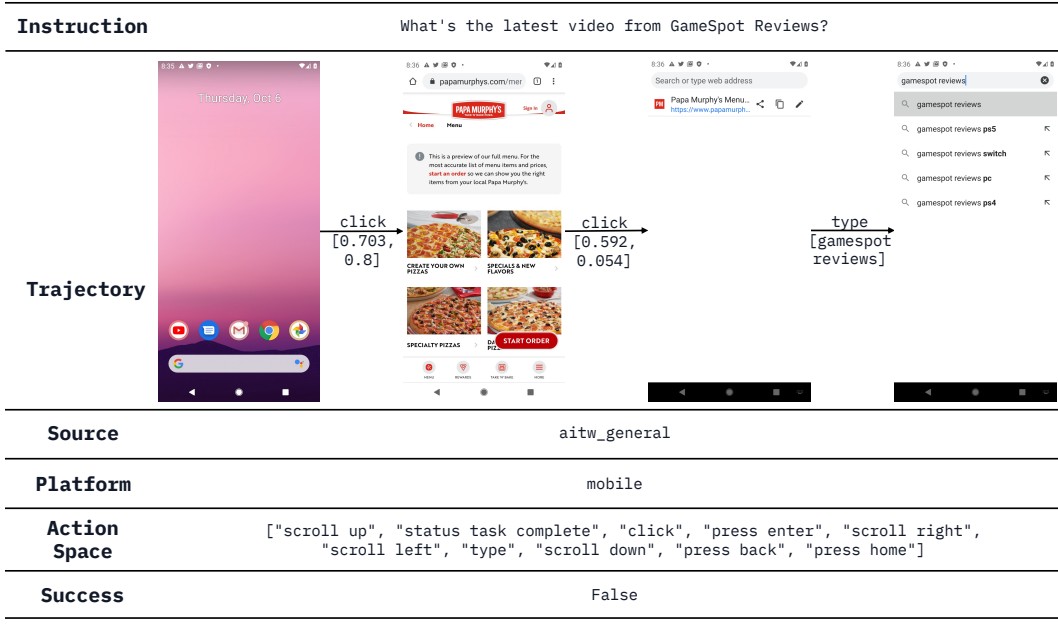

| Instruction | What's the latest video from GameSpot Reviews? |
|---|---|
| Trajectory | |
| Source | aitw_general |
| Platform | mobile |
| Action Space | ["scroll up", "status task complete", "click", "press enter", "scroll right", "scroll left", "type", "scroll down", "press back", "press home"] |
| Success | False |

Figure 16: Data sample on mobile platform in IDM-Multiple dataset.

| Instruction | apply for a delivery driving job at Grand Rapids, MI, USA. Name: Nelson Freeman, Email: Nelsonfree@gmail.com, phone: 2034556656. Select \"yes\" for all options |
|---|---|

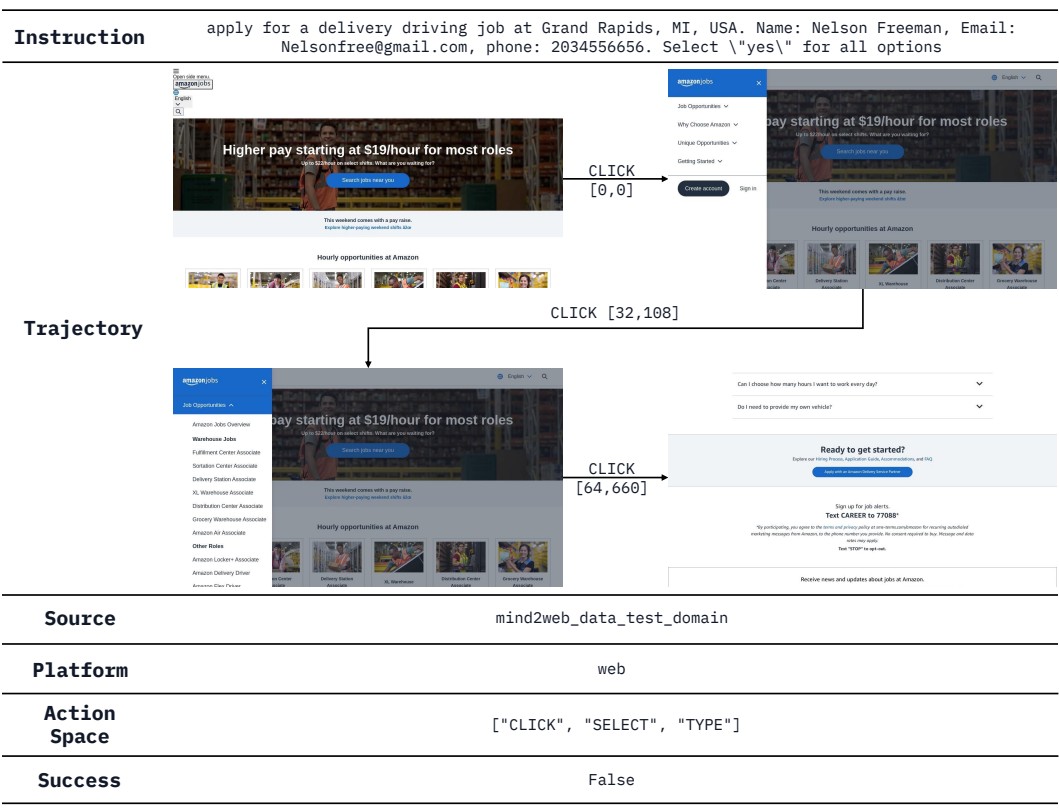

| Trajectory | |

| Source | mind2web_data_test_domain |
|---|---|
| Platform | web |
| Action Space | ["CLICK", "SELECT", "TYPE"] |
| Success | False |

Figure 17: Data sample on web in IDM-Multiple dataset.

## D.2    PROMPTS FOR IDM-SINGLE

The prompt for IDM-Single:

---

**User**

`<datasets/trajectory_lite/images/aitw_images/googleapps/`
`11660885229547098410_3.png>`

`<datasets/trajectory_lite/images/aitw_images/googleapps/`
`11660885229547098410_4.png>`

Analyze the two sequential images provided. These images represent observations before and after an action was taken. Your task is to identify which type of action was executed between these two observations.
Available actions:
A. scroll up
B. status task complete
C. click
D. press enter
E. scroll right
F. scroll left
G. type
H. scroll down
I. press back
J. press home
After your analysis, conclude your response with the answer in this format:
Answer: X
Where X is the letter corresponding to the action you believe was taken, chosen from the options provided above.
For example, if the options were:
A: Type
B: Click
C: Scroll
And you determined the action was "Click", your response would end with:
Answer: B

---

**Assistant**
Analyzing the two images, we can observe the following changes:
1. In the first image, we see a partial list of settings including "Bluetooth", "Apps & notifications", "Battery", and "Display".
2. In the second image, the list has moved upwards, revealing additional settings below "Display", namely "Sound" and "Storage".
3. The top items from the first image ("Bluetooth" and "Apps & notifications") are no longer visible in the second image.
4. The battery icon and percentage (100%) remain constant between the two images.
5. The time (2:12) and status bar icons at the top of the screen remain the same.
These observations indicate that the screen content has moved upwards, revealing more options at the bottom while hiding options that were previously at the top.
Given these changes, the action that was most likely taken between these two observations is scrolling down on the settings menu.
Answer: H

---

**Evaluation**
AgentStudio will parse the response into H, which is the same as the ground truth action. Therefore, the score is 1.0.

---

## D.3 PROMPTS FOR IDM-MULTIPLE

For IDM-Multiple, we prompt with up to six consecutive observations from the sampled trajectories.

---

**User**

```
<datasets/trajectory_lite/images/aitw_images/install/
803558926902382084_0.png>

<datasets/trajectory_lite/images/aitw_images/install/
803558926902382084_1.png>

<datasets/trajectory_lite/images/aitw_images/install/
803558926902382084_2.png>

<datasets/trajectory_lite/images/aitw_images/install/
803558926902382084_3.png>

<datasets/trajectory_lite/images/aitw_images/install/
803558926902382084_4.png>

<datasets/trajectory_lite/images/aitw_images/install/
803558926902382084_5.png>
```

Analyze the sequence of images provided. These images are snapshots taken from a video screen recording.
Your task is to identify the types of actions executed in order throughout this recording.
Available actions:
A. scroll up
B. status task complete
C. click
D. press enter
E. scroll right
F. scroll left
G. type
H. scroll down
I. press back
J. press home
After your analysis, conclude your response with the answer in this format:
Answer:
X
Y
Z
...
Where X, Y, Z, etc. are letters corresponding to the actions you believe were taken, chosen from the
options provided above.
For example, if the options were:
A: Type
B: Click
C: Scroll
And you determined the actions were "Type", then "Scroll", your response would end with:
Answer:
A
C

---

**Assistant**
Based on analyzing the sequence of images, here's my interpretation of the actions taken:
1. The first image shows the home screen.
2. The second image shows a search or app drawer opened.
3. The third image displays the Google Play Store page for the Contacts app.
4. The fourth image shows a search results page for "contacts" within an app store or system.
5. The fifth image displays search results for apps and games.
6. The last image shows search results for "YouTube Kids".
Given this sequence, here's my assessment of the actions taken:
1. The user clicked to open the app drawer or search (C)
2. They typed "contacts" to search (G)
3. They scrolled down to view more results (H)
4. They clicked on the Contacts app result (C)
5. They pressed back to return to the search (I)
6. They typed to modify the search query, likely changing it to "YouTube Kids" (G)
Answer:
C
G
H
C
I
G

**Evaluation**
AgentStudio will parse the response into (C, G, H, C, I, G). The correct action sequence is (C, C, C, C, G).
Each letter represents an available action in the action space. Therefore, the model generates an incorrect
action sequence. The predicted action sequence requires at least three times of modification (delete, insert,
or replace) to become the correct one. Therefore, the score is 0.0, and the edit distance is 3.

# E DETAILS OF CRITICBENCH

Figure 18 and Figure 19 are two examples from CriticBench.

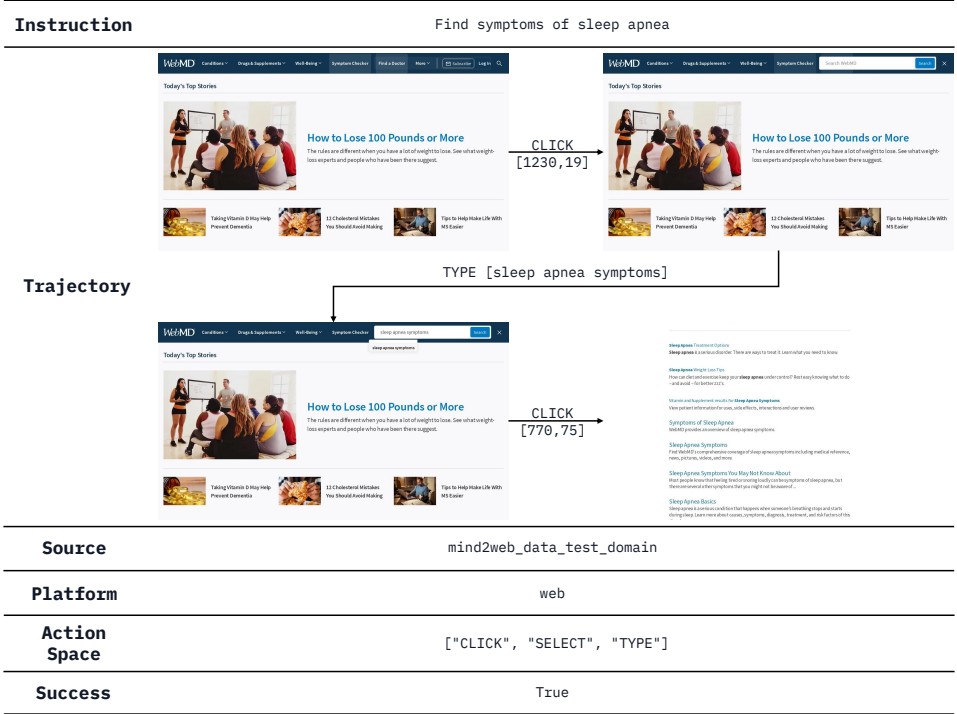

Figure 18: Data sample on web in CriticBench dataset.

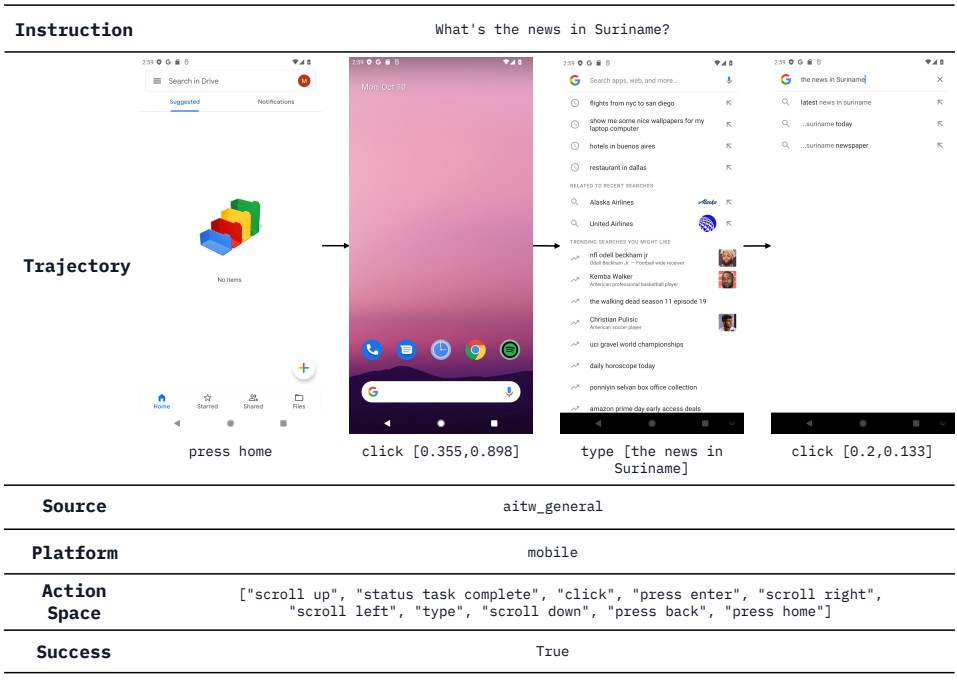

Figure 19: Data sample on mobile platform in CriticBench dataset.

### E.1 PROMPTS FOR CRITICBENCH WITH OBSERVATION-ACTION PAIRS

**User**

`<datasets/trajectory_lite/images/aitw_images/general/`
`15667803399207554024_0.png>`

press home

`<datasets/trajectory_lite/images/aitw_images/general/`
`15667803399207554024_1.png>`

click NAV BAR CIRCLE

`<datasets/trajectory_lite/images/aitw_images/general/`
`15667803399207554024_2.png>`

type the news in Suriname

`<datasets/trajectory_lite/images/aitw_images/general/`
`15667803399207554024_3.png>`

click MAGNIFYING GLASS
Analyze the sequence of images provided. These images are snapshots taken from a video screen recording. Your task is to evaluate whether the given instruction has been successfully completed based on the provided trajectory. This trajectory is depicted through a series of actions and the corresponding observations before and after each action.
Instruction: What's the news in Suriname?
Action space: scroll up, status task complete, click, press enter, scroll right, scroll left, type, scroll down, press back, press home
Carefully examine the sequence of actions and their results. Determine if the final state successfully fulfills the given instruction.
Conclude your analysis with one of these two responses:
If the trajectory successfully completes the instruction:
Answer: True
If the trajectory fails to complete the instruction:
Answer: False

**Assistant**
Let's analyze the sequence of images and actions:
1. The first image shows the Google Drive app interface.
2. The second image displays the Android home screen.
3. The third image shows the Google search bar with recent searches and trending topics.
4. The fourth image shows the Google search bar with "the news in Suriname" typed in, along with search suggestions related to Suriname news.
Based on this sequence, we can see that:
1. The user navigated from Google Drive to the home screen.
2. They then accessed the Google search bar.
3. They typed "the news in Suriname" into the search bar.
However, the sequence ends before actually searching for or displaying any news from Suriname. The user has only typed the query, but hasn't pressed enter or clicked on any search results to view the actual news. Therefore, while the user has taken steps towards finding news in Suriname, they haven't actually accessed or viewed any news content yet.
Conclusion:
Answer: False
The trajectory does not fully complete the instruction of finding out what the news in Suriname is. It only reaches the point of typing the search query, without actually retrieving or displaying any news content.

**Evaluation**
AgentStudio will parse the response to get the answer "False". However, the given trajectory successfully completes the instruction. Therefore, the score is 0.0.

## E.2 PROMPTS FOR CRITICBENCH WITH OBSERVATIONS ONLY

**User**

`<datasets/trajectory_lite/images/aitw_images/general/5055803312245834626_8.png>`

`<datasets/trajectory_lite/images/aitw_images/general/5055803312245834626_9.png>`

`<datasets/trajectory_lite/images/aitw_images/general/5055803312245834626_10.png>`

`<datasets/trajectory_lite/images/aitw_images/general/5055803312245834626_11.png>`

`<datasets/trajectory_lite/images/aitw_images/general/5055803312245834626_12.png>`

Analyze the sequence of images provided. These images are snapshots taken from a video screen recording. Your task is to evaluate whether the given instruction has been successfully completed based on the provided images.
Instruction: Check the settings for the Amazon Alexa app
Carefully examine the sequence of images. Determine if the final state successfully fulfills the given instruction.
Conclude your analysis with one of these two responses:
If the trajectory successfully completes the instruction:
Answer: True
If the trajectory fails to complete the instruction:
Answer: False

**Assistant**
After carefully examining the sequence of images provided, I can conclude:
Answer: False
Reasoning:
1. The images show a smartphone home screen with a pink gradient background.
2. The date, time, and weather are displayed at the top.
3. There are several app icons visible at the bottom of the screen, including Google Play, Messages, Clock, Chrome, and Gmail.
4. A Google search bar is present at the bottom.
5. In some images, there's a "My apps" menu partially visible.
However, throughout the sequence, there is no indication that the Amazon Alexa app is opened or that its settings are being checked. The instruction was to "Check the settings for the Amazon Alexa app," but the images do not show any interaction with the Alexa app or its settings. Therefore, the given instruction has not been successfully completed based on the provided images.

**Evaluation**
AgentStudio will parse the response to get the answer "False". However, the given trajectory successfully completes the instruction. Therefore, the score is 0.0.

# F    DETAILS OF AGENTSTUDIO BENCHMARK TASKS

## F.1    SYSTEM PROMPT

**System**
You are a world-class programmer who can complete any instruction by executing Python code. Now you are operating a real computer-based environment, and you may be given a screenshot of the current computer screen. The only way to interact with the environment is to write Python code.
You are given a task instruction in the form of a string and you need to write Python code to complete it. You are using Jupyter Notebook to execute the code and shell commands, so generate code/shell commands step by step with multiple blocks. You can use Jupyter internal operators "%" and "!". The generated code/shell commands should be wrapped between "```python\n" and "\n```". Your response should include and only include one code block. You will get the execution result of the code.
You can interact with the Notebook multiple rounds. Thus, if you are not sure about the code, you can submit the code to see the result and modify the code if needed. When you think the instruction is completed and the code is correct, end the code block with 'exit()'.
For simplicity, you can use the following code snippets:
You are probably given a screenshot of the current computer screen. You can only use the Jupyter Notebook to interact with the environment. We have provided the initial code to access the mouse and keyboard:

```
from agent_studio.envs.desktop_env import Mouse, Keyboard
mouse = Mouse()
keyboard = Keyboard()
```

You can use the 'mouse' and 'keyboard' objects to interact with the environment. 'mouse.click(x:int,y:int,button:str,clicks:int,interval:float)' can be used to click the "button" at the specified position "click" times with a specific interval. You can choose "button" from "left", "right", and middle. 'keyboard.type(text:str,interval:float)' can be used to type the specified text with a specific interval. 'keyboard.hotkey(keys:list[str])' can be used to press hotkeys.
If your task needs to access the Google service, you can use the 'credentials.json' file in the './agent_studio/config' directory. Also, there are six token files, 'docs_token.json', 'drive_token.json', 'gmail_token.json', 'sheets_token.json', 'slides_token.json', 'calendar_token.json' and 'forms_token.json', in the './agent_studio/config' directory, and you can use any of them to access the corresponding Google service. E.g. you can use the following code to access the Google Drive API:

```
import json
from google.oauth2 import credentials
from googleapiclient.discovery import build

token_path="agent_studio/config/docs_token.json"
with open(token_path, "r") as f:
    token = json.loads(f.read())
creds = credentials.Credentials.from_authorized_user_info(
    token, ["https://www.googleapis.com/auth/drive",]
)
service = build("drive", "v3", credentials=creds)
service.files().get_media(fileId=xxxxxxx)
```

Also, you should assume the timezone is UTC+0 if there's no further specification.

## F.2    PROMPTS FOR TASKS WITH TEXT OBSERVATIONS ONLY

**System**
```
<SYSTEM PROMPT in Appendix F.1>
```

**User**
The task instruction: `<TASK SPECIFIC INSTRUCTION>`

**Assistant**
Action: `<PYTHON CODE 1>`

**User**
Observation: `<PYTHON EXECUTION RESULT 1>`

**Assistant**
Action: `<PYTHON CODE 2>`

**User**
Observation: `<PYTHON EXECUTION RESULT 2>`

## F.3  PROMPTS FOR TASKS WITH TEXT AND VISUAL OBSERVATIONS

**System**
`<SYSTEM PROMPT in Appendix F.1>`

**User**
The task instruction: `<TASK SPECIFIC INSTRUCTION>`

**Assistant**
Action: `<PYTHON CODE 1>`

**User**
Observation: `<PYTHON EXECUTION RESULT 1>`

**Assistant**
Action: `<PYTHON CODE 2>`

**User**
Observation: (`<SCREENSHOT>`, `<PYTHON EXECUTION RESULT 2>`)

## F.4 EXAMPLE TASK CONFIGURATION

| Key | Value |
|---|---|
| Task ID | 08aced46-45a2-48d7-993b-ed3fb5b32302 |
| Instruction | Give the slide 2 a right aligned title, "Note". |
| Visual | True |
| Max Steps | 30 |
| Max Time | 60.0 |
| Evaluation Procedure | Compare between `ref.pptx` and `target.pptx` |
| Reset Procedure | 1. Create folder structure, 2. Copy file, 3. Open PPTX file |
| Cleanup Procedure | 1. Delete folder structure, 2.Kill LibreOffice process |

Table 12: Task configuration with simplified evaluation/reset/cleanup procedures.

## F.5 EXAMPLE FAILED TRAJECTORY

**Task ID**: 0a2e43bf-b26c-4631-a966-af9dfa12c9e5

**Instruction**: Work out the monthly total sales in a new row called "Total" and then create a line chart to show the results (x-axis be Months).

**Model**: GPT-4o (0806)

> **User**
> The task instruction: Work out the monthly total sales in a new row called "Total" and then create a line chart to show the results (x-axis be Months).

> **Assistant**
> To complete this task, we need to perform the following steps:
> 1. Calculate the total sales for each month and add a new row called "Total".
> 2. Create a line chart to display the monthly total sales.
> Let's proceed with the code to achieve this.
>
> ```python
> import pandas as pd
> import matplotlib.pyplot as plt
>
> # Load the Excel file
> file_path = 'SalesRep.xlsx'
> df = pd.read_excel(file_path, sheet_name='Sheet1')
>
> # Calculate the total sales for each month
> df.loc['Total'] = df.sum(numeric_only=True)
>
> # Plot the line chart
> plt.figure(figsize=(10, 6))
> plt.plot(df.columns[1:], df.loc['Total'][1:], marker='o')
> plt.title('Monthly Total Sales')
> plt.xlabel('Months')
> plt.ylabel('Total Sales')
> plt.grid(True)
> plt.show()
> ```

**User**
Observation:
Screenshot

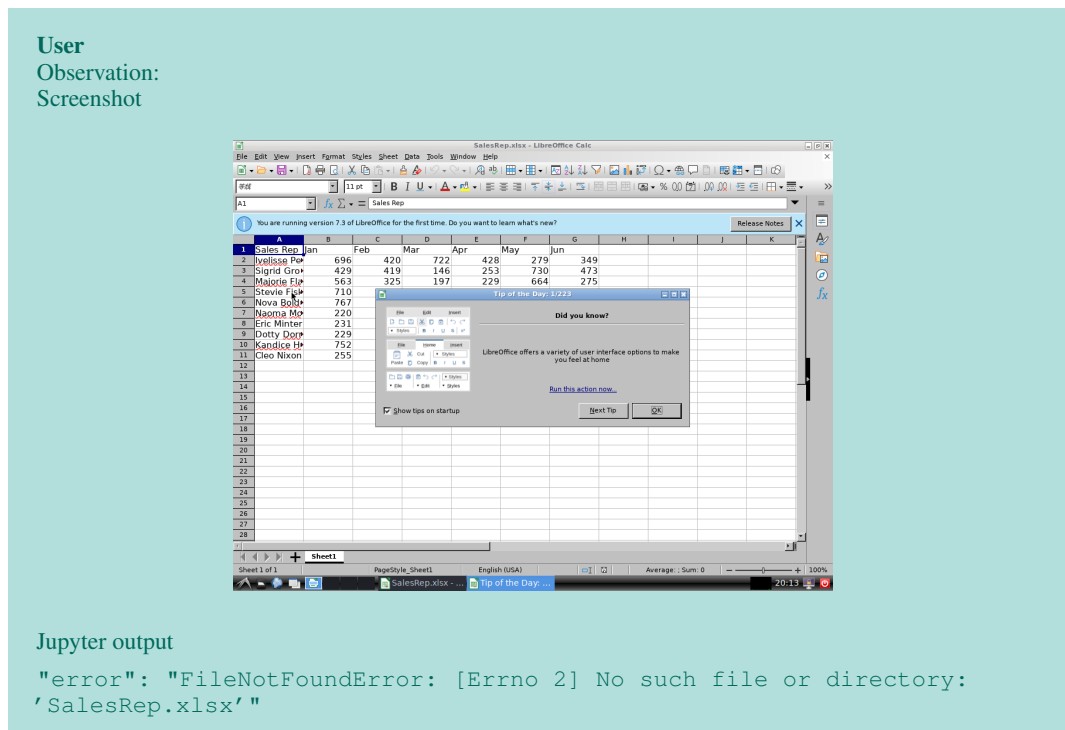

Jupyter output

```
"error": "FileNotFoundError: [Errno 2] No such file or directory:
'SalesRep.xlsx'"
```

**Assistant**
To calculate the monthly total sales and create a line chart, follow these steps:
1. Close the "Tip of the Day" dialog.
2. Add a new row for the total sales.
3. Use a formula to sum each month's sales.
4. Create a line chart.
Let's proceed with the steps using Python code to automate this process.

```python
from agent_studio.envs.desktop_env import Mouse, Keyboard
import time

mouse = Mouse()
keyboard = Keyboard()

# Close the "Tip of the Day" dialog
mouse.click(x=600, y=400, button='left', clicks=1, interval=0.1)
time.sleep(1)

# Click on the cell to enter the total formula for January
mouse.click(x=200, y=300, button='left', clicks=1, interval=0.1)
keyboard.type('=SUM(B2:B11)', interval=0.1)
keyboard.hotkey(['Enter'])
time.sleep(1)

# Repeat for the other months
columns = ['C', 'D', 'E', 'F', 'G', 'H']
for i, col in enumerate(columns, start=3):
    mouse.click(
        x=200 + (i-2)*100,
        y=300,
        button='left',
        clicks=1,
        interval=0.1
    )
    keyboard.type(f'=SUM({col}2:{col}11)', interval=0.1)
    keyboard.hotkey(['Enter'])
    time.sleep(1)

# Create a line chart
# Click on the Insert menu
mouse.click(x=50, y=50, button='left', clicks=1, interval=0.1)
time.sleep(1)

# Click on Chart
mouse.click(x=100, y=200, button='left', clicks=1, interval=0.1)
time.sleep(1)

# Follow the chart wizard steps to create a line chart
# This part may require manual adjustments depending on the interface

exit()
```

**User**
Observation:
Screenshot

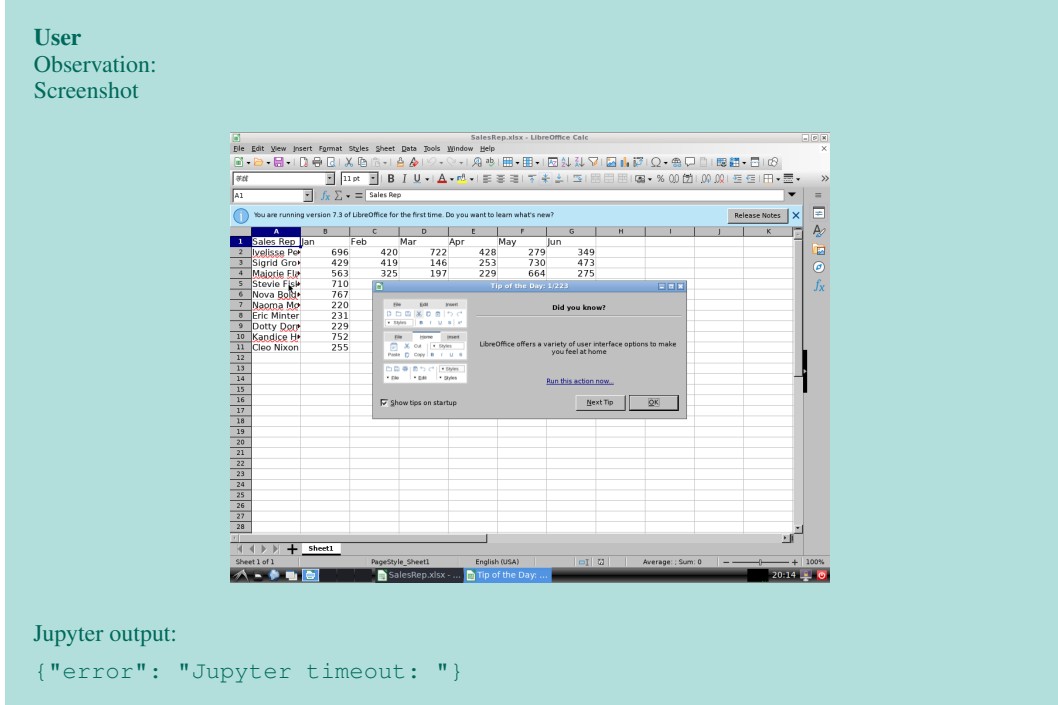

Jupyter output:
```
{"error": "Jupyter timeout: "}
```

Exit because of: Agent outputs 'exit()'

**Evaluator**
score: 0.0
feedback: "rule sheet_data failed, sheet data matching failed"

## F.6 Example Successful Trajectory

**Task ID**: 3dcc6db1-7ca5-412d-b519-142724d41ef2

**Instruction**:
Open VSCode with the file "/home/ubuntu/agent_studio/vscode_workspace/README.md".

**Model**: GPT-4o (0806)

**User**
The task instruction:
Open VSCode with the file "/home/ubuntu/agent_studio/vscode_workspace/README.md".
Screenshot:

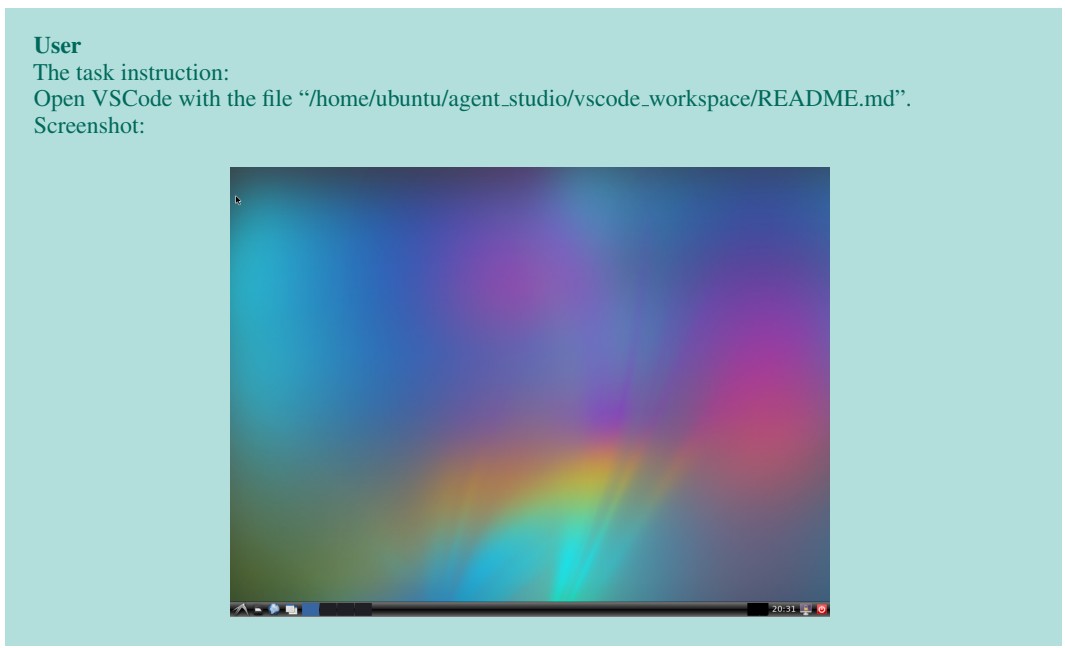

**Assistant**
To open VSCode with the specified file, I'll execute the command to launch VSCode from the terminal.

```python
!code /home/ubuntu/agent_studio/vscode_workspace/README.md
```

**User**
Screenshot:

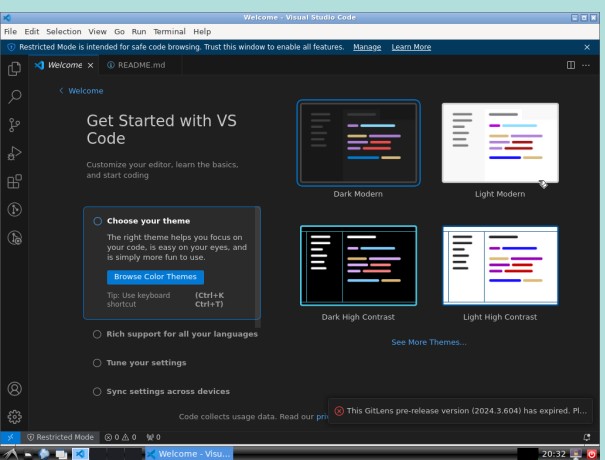

Jupyter output
```
No output
```

**Assistant**
It looks like VSCode is open with the README.md file. If you need further assistance or want to perform another action, let me know!

Exit because of: No valid action

**Evaluator**
score: 1.0

## F.7 ADDITIONAL EXAMPLES

| Task ID | 581aab2a-57d5-4384-841f-2f7655491859 | 1e2f5ea6-c448-48e9-a2d1-be6f5639b4b2 |
|---|---|---|
| Observation | Text | Text & Screenshots |
| Instruction | Delete the file 'temporary.docx' in the '/home/ubuntu/agent_studio/tmp/docs/' directory. | Delete the file 'temporary.docx' in the '/home/ubuntu/agent_studio/tmp/docs/' directory. |
| Response 1 | To delete the file 'temporary.docx' in the specified directory, I'll execute the necessary shell command.
`!rm /home/ubuntu/agent_studio/tmp/docs/temporary.docx` | ```import os # Define the file path file_path = '/home/ubuntu/agent_studio/ tmp/docs/temporary.docx' # Check if the file exists if os.path.exists(file_path):     # Delete the file     os.remove(file_path)     print(f"File 'file_path'     has been deleted.") else:     print(f"File 'file_path'     does not exist.")``` |
| Action 1 | `!rm /home/ubuntu/agent_studio/tmp/docs/temporary.docx` | ```import os # Define the file path file_path = '/home/ubuntu/agent_studio/ tmp/docs/temporary.docx' # Check if the file exists if os.path.exists(file_path):     # Delete the file     os.remove(file_path)     print(f"File 'file_path'     has been deleted.") else:     print(f"File 'file_path'     does not exist.")``` |
| Response 2 | It seems you have provided a screenshot of a desktop environment. If you need assistance with a task related to this environment, please let me know how I can help! | The file 'temporary.docx' has been successfully deleted from the '/home/ubuntu/agent_studio/tmp/docs/' directory. If you have any further tasks or need additional assistance, feel free to ask! |
| Action 2 | None | None |

Table 13: Additional comparisons between GUI and non-GUI online benchmark tasks.

## F.8 SOFTWARE SELECTION

Our software selection is based on systematic criteria. (1) Open-source: We select open-source software to avoid copyright issues and encourage community contributions. (2) Popularity: We include widely used tools, such as Google Workspace, VS Code, and LibreOffice, which are among the most prevalent applications in their respective categories. GIMP (GNU Image Manipulation Program) is one of the most popular open-source image editors. (3) Diversity: To represent a broad range of real-world use cases, we select software across different domains, including OS components, office suites, code editors, and image editors. Although our environment can support proprietary software like Photoshop, we prioritize using ones that do not require licenses for running benchmarks. Meanwhile, our benchmark tasks for GIMP are designed to be easily applicable to Photoshop as well, since most image-related tasks are evaluated by checking the output images.

## F.9 MORE TECHNICAL DETAILS

Our implementation uses Docker for a lightweight environment compared to virtual machines used by OSWorld (Xie et al., 2024). Key technical aspects include: (1) Performance & latency: The runtime

mainly depends on the time for task reset, action generation & execution, auto-eval, and cleanup. AgentStudio is designed to mimic real-world computers operating in real time. The latency of reset, auto-eval, and cleanup depends on the Docker resource configuration, which can be improved by allocating additional resources to Docker. At present, the main bottleneck is the response time of LLM action generation, as well as the time required to execute LLM-generated code, rather than the latency in AgentStudio. (2) Scalability: Our architecture supports multiple instances running concurrently. The separation of the front end and back end ensures that LLM operations within Docker containers do not interfere with each other. Google API operations can also be isolated by setting up different accounts. (3) Reliability: Our Docker setup safeguards local files and systems from potential vulnerabilities. For sensitive operations, AgentStudio supports enabling human verification to enhance reliability and safety.

| Instruction | Update the content of 'agenda.txt' in '/home/ubuntu/agent_studio/tmp/meetings/' to 'Updated Meeting Agenda'. |
|---|---|
| Non-visual | ```
from agent_studio.envs.desktop_env import Mouse, Keyboard
mouse = Mouse()
keyboard = Keyboard()
with open(
    '/home/ubuntu/agent_studio/tmp/meetings/agenda.txt',
    'w'
) as f:
    f.write('Updated Meeting Agenda')
    exit()
``` |
| Visual | ```
from agent_studio.envs.desktop_env import Mouse, Keyboard
mouse = Mouse()
keyboard = Keyboard()
keyboard.hotkey(['super'])
keyboard.type('
    gedit /home/ubuntu/agent_studio/tmp/meetings/
    agenda.txt'
)
keyboard.hotkey(['enter'])
mouse.click(900, 500, 'left', 1, 0.1)
keyboard.hotkey(['ctrl', 'a'])
keyboard.type('Updated Meeting Agenda')
mouse.click(20, 20, 'left', 1, 0.1)
mouse.click(15, 20, 'left', 1, 0.1)
exit()
``` |

Table 14: Actions of Gemini 1.5 Pro in visual and non-visual tasks. When providing visual information (screenshot), the Gemini model is more likely to be distracted by the visual information. It will try to use keyboard and mouse operations to complete the task, no matter how easy the problem is solved with the API.

# G  DETAILS OF AGENTSTUDIO TOOLS

## G.1  GUI ANNOTATION

To use the tool for GUI element annotation, first, click the "Capture" button to take a screenshot, which will appear on the left side of the interface. After the screenshot is captured, input the annotation instruction in the text box labeled "Enter instruction here." Then, draw a bounding box around the relevant area in the screenshot to match the instruction. Once the instruction and bounding box are set, click "Save" to store the annotation. If users need to start over, click "Reset" to clear the screenshot and instructions, then repeat the process as needed for more annotations.

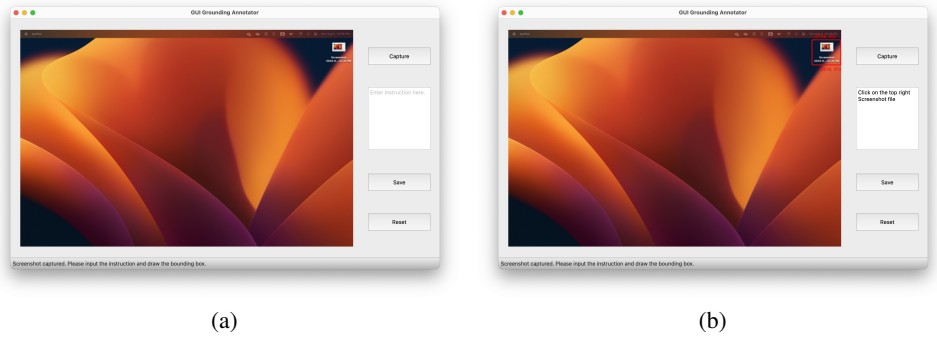

|(a)|(b)|

Figure 20: (a) Step 1: Capture a screenshot. (b) Step 2: Input instruction and draw the bounding box.

## G.2  ACTION RECORDING

The trajectory recorder is a Command Line Interface (CLI) tool designed to capture mouse actions, keyboard inputs, and screen recordings. As shown in Table 15, to use the tool, begin by selecting a path where the recording will be saved and then input an instruction describing the task users intend to complete. The recorder will initiate a 5-second countdown, after which users will hear a chime signaling the start of the recording. Once users have completed the task, stop the recording by pressing the ctrl+shift+o key combination.

```
$ as-traj-recorder
Please enter the folder path where you want to save the record
(Press Enter to use the default path: ~/home/ubuntu):

Please input the instruction:
> Please help me open the autosave feature of VS Code and delay
AutoSave operations for 500 milliseconds in the VS Code setting.

recording will start in 5 seconds
You will hear a sound when recording starts
5
4
3
2
1
press ctrl+shift+o to stop recording
```

Table 15: Illustration of the trajectory recorder in use.

### G.3 VIDEO REFINEMENT

Figure 21 illustrates the user interface of our trajectory editor. Users can open a recording saved with the tool in Appendix G.2 and view the keyboard or mouse operations in the right list view. Here, users can leverage shortcuts to instantly modify the recorded operations, e.g., deleting an operation, aggregating several operations, and changing operation information.

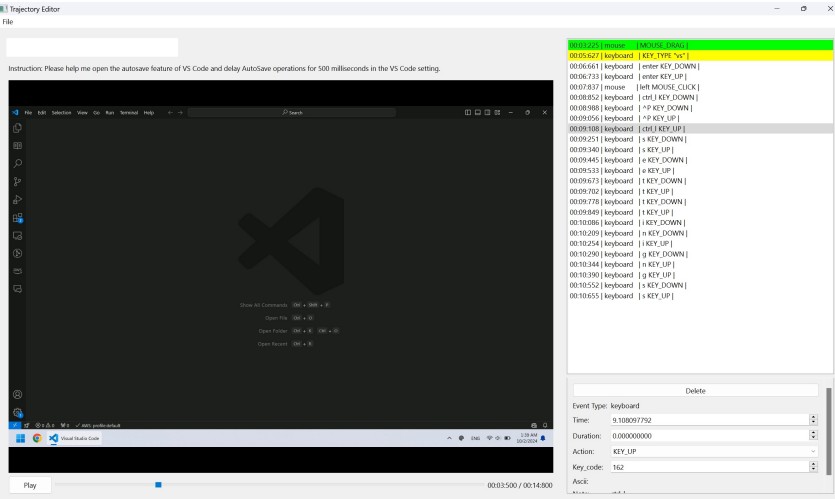

Figure 21: The interface for video refinement.

### G.4 ONLINE BENCHMARK VISUALIZATION & HUMAN VALIDATION

Figure 22 shows the GUI of our online benchmark, where users can select and view task configurations, execute the task, view the action execution results, and get feedback. Users can leverage this interface to evaluate a task and calculate human completion time.

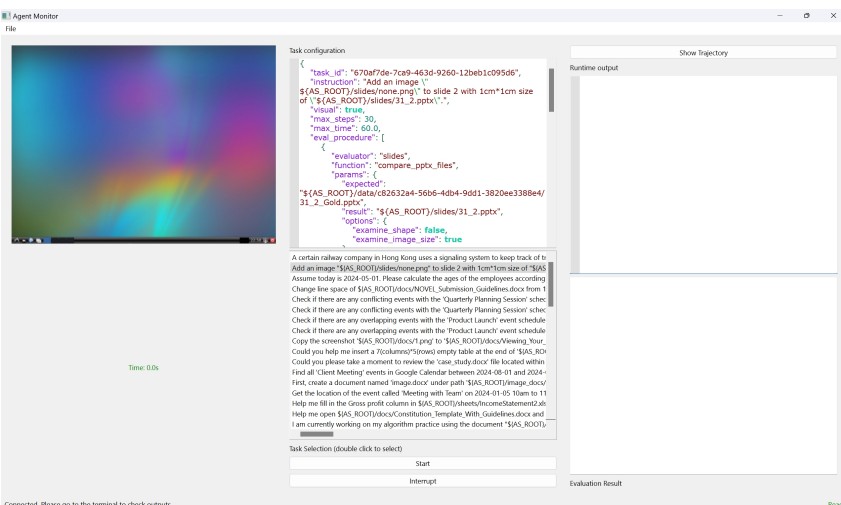

Figure 22: Online benchmark in GUI mode.

