# OpenReview forum: "AgentStudio: A Toolkit for Building General Virtual Agents"
_ICLR.cc/2025/Conference — ICLR 2025 Poster_

### Official Review · Reviewer_iPnp · 2024-11-02

**Soundness:** 4
**Presentation:** 3
**Contribution:** 3
**Rating:** 8
**Confidence:** 2

**Summary:**

This paper presents a comprehensive toolkit that includes an environment, tools, benchmarks, and datasets designed to evaluate virtual agents’ capabilities across diverse software tasks. The authors focus on enabling realistic evaluations of agents in real-world settings, providing tools to easily construct these environments within a POMDP framework. They conduct extensive experiments on their benchmark, covering various GUI and API-related tasks, to assess the overall performance of current agents and their effectiveness in three specific tasks. Additionally, they introduce three datasets that address fundamental challenges, highlighting limitations in current models.

**Strengths:**

1. AgentStudio provides a holistic toolkit, including environments, tools, and benchmarks, addressing the need for flexible agent training across varied virtual scenarios​. It also supports diverse input modalities (text, images, videos) and action types (GUI, APIs), making it versatile for training agents to handle real-world tasks and improving generalizability.
2. The creation of GroundUI, IDMBench, and CriticBench highlights an effort to evaluate core agent skills such as GUI grounding and success detection, advancing the field with nuanced benchmarks​.
3. AgentStudio's lightweight design and compatibility with various operating systems (including Docker environments) improve accessibility and ease of use for a wider research audience​.

**Weaknesses:**

1. Although AgentStudio emphasizes versatility and real-world applicability, the benchmarks and tasks are still simulated in controlled environments, which may not fully capture the complexities of real-world application. The challenges faced by agents in uncontrolled, dynamic environments are difficult to replicate, which could limit the generalizability of the findings to actual user settings.
2. Success rates and accuracy are used as primary metrics, which may not fully capture an agent's capabilities, particularly in scenarios requiring complex decision-making or nuanced interaction. For example, success rates don’t account for partial progress in multi-step tasks, which may lead to an incomplete assessment of an agent’s performance and fail to identify areas where improvement is needed.

**Questions:**

See above Weakness

---

> ### Author Response · Authors · 2024-11-18
> **Response to Reviewer iPnp**
>
> Thank you for your valuable feedback. Below, we address each of your points in detail.
>
> > **W1**: Although AgentStudio emphasizes versatility and real-world applicability, the benchmarks and tasks are still simulated in controlled environments, which may not fully capture the complexities of real-world application. The challenges faced by agents in uncontrolled, dynamic environments are difficult to replicate, which could limit the generalizability of the findings to actual user settings.
>
> Real-world applications often involve more complexity. Therefore, we use controlled settings to create benchmark tasks to keep them lightweight and reproducible. At the same time, AgentStudio environment and tools are fully compatible with other real-world applications. This design choice ensures the reproducibility of benchmarks while preserving the flexibility to scale them to real-world settings.
>
> > **W2**: Success rates and accuracy are used as primary metrics, which may not fully capture an agent's capabilities, particularly in scenarios requiring complex decision-making or nuanced interaction. For example, success rates don’t account for partial progress in multi-step tasks, which may lead to an incomplete assessment of an agent’s performance and fail to identify areas where improvement is needed.
>
> We acknowledge that success rates and accuracy are not the only metrics for evaluating agents. While it is challenging to assign scalar rewards for partial progress in multi-step tasks, and the common practice is to use binary rewards [1, 2, 3], AgentStudio offers more detailed feedback beyond binary rewards. Specifically, for compositional tasks involving multiple auto-evaluators assessing task outcomes, AgentStudio can highlight which evaluators passed or failed and provide language feedback for these outcomes, enabling both humans and agents to analyze and improve agent performance.
>
> [1] Xie, Tianbao, et al. "OSWorld: Benchmarking multimodal agents for open-ended tasks in real computer environments." NeurIPS 2024 Datasets and Benchmarks Track.
>
> [2] Zhou, Shuyan, et al. "WebArena: A realistic web environment for building autonomous agents." ICLR 2024.
>
> [3] Koh, Jing Yu, et al. "VisualWebArena: Evaluating multimodal agents on realistic visual web tasks." ACL 2024.

---

### Official Review · Reviewer_uDrN · 2024-11-03

**Soundness:** 2
**Presentation:** 3
**Contribution:** 2
**Rating:** 6
**Confidence:** 5

**Summary:**

This paper presents AgentStudio, a toolkit for building and evaluating general virtual agents that can interact with software through GUI and API interfaces. The main contributions are: (1) An interactive environment supporting multimodal observations and actions, (2) Tools for creating benchmark tasks and annotating data, (3) A benchmark suite of 205 tasks and three datasets (GroundUI, IDMBench, CriticBench) for evaluating fundamental agent capabilities.

**Strengths:**

1. The interactive environment design allowing both GUI and API interactions is valuable

2. It introduces an online task-completion benchmark and three datasets to evaluate fundamental agent abilities in real-world settings. The benchmark suite consists of 205 real-world tasks across various applications such as VS Code, Google Workspace,
and Office suites

3. Evaluating current LLM-based agents (Claude 3.5 Sonnet, GPT-4o, Gemini 1.5 Qwen-VL-Chat) on real-world software interaction tasks, it provides good analysis of failure modes and limitations of existing models.

**Weaknesses:**

1. Limited technical novelty - the environment appears to be largely an integration of existing components without significant new technical contributions.

2. The three datasets created for fine-grained evaluation are quite small in scale: IDMBench, criticBench has only 345, 350 trajectories respectively.

3. Insufficient technical details about the implementation:
Only cursory mention of using VNC and Docker
No discussion of performance, latency, scalability, reliability considerations

**Questions:**

Can AgentStudio enable large scale data collection, and agent evaluation? Will it support multi-threading that can be integrated to online RL training of agents?

What are the key technical differences between AgentStudio and existing environments like VisualWebArena?

Why are the evaluation datasets so small? Are there plans to scale them up?

In IDM-Multiple, accuracy is calculated based on the exact match of the action sequences. Is this too restrictive? Would there be multiple optimal action sequences? Even suboptimal, but successful sequences demonstrate agent capabilities.

The paper presents an interesting system but falls short in terms of technical novelty and scale of evaluation. The main contribution appears to be integration of existing components rather than fundamental technical advances. The small scale of the evaluation datasets also limits the impact. While the direction is promising, the current work would benefit from more technical depth and larger-scale evaluation.

Post rebuttal:
There are still further questions on how useful AgentStudio is as many components are very basic, e.g. annotation tool as characterized "minimalism". Furthermore, it is not clear college students are typical users and whether there are any quality control processes, the 51% calls into questions to the labeling process.

I still feel the paper is borderline. Nonetheless, I am happy to update my score.

---

> ### Author Response · Authors · 2024-11-18
> **Response to Reviewer uDrN (1/2)**
>
> Thank you for your valuable feedback. Below, we address each of your points in detail.
>
> > **W1**: Limited technical novelty - the environment appears to be largely an integration of existing components without significant new technical contributions.
>
> As detailed in the introduction and related work sections, AgentStudio offers several key advantages over existing solutions, which is also acknowledged by Reviewer v3Q2. Specifically:
> 1. **Environment**: Our environment is more lightweight, simulates the complexity of real-time computers, and supports general observation and action spaces, including video observations and API+GUI actions. This versatility is broader than previous works, facilitating more general agents.
> 2. **Benchmarks**: We offer online benchmark tasks alongside three specialized datasets to evaluate fundamental agent abilities. The key contributions are:
>    - **Online tasks**: These tasks incorporate GUI and API actions, presenting new research opportunities such as selecting between different action spaces.
>    - **Datasets decomposing agent abilities**: We identify three desiderata for general virtual agents and provide detailed evaluations. Currently, there are no benchmarks similar to IDMBench and CriticBench. Additionally, GroundUI offers a systematically curated UI grounding dataset in a unified format across various apps and devices.
> 3. **Tools**: We also provide tools for data collection, annotation, and benchmark creation, which helps expand benchmarks and scale up training data.
>
> > **W2**: The three datasets created for fine-grained evaluation are quite small in scale: IDMBench, criticBench has only 345, 350 trajectories respectively.
>
> We intentionally selected a representative subset of our trajectory datasets to ensure efficient and cost-effective evaluations for researchers. Since each trajectory contains multiple images, IDMBench and CriticBench are smaller than GroundUI to maintain a similar evaluation token cost. Notably, we have a larger trajectory dataset that could be utilized to scale up IDMBench and CriticBench. The original trajectory datasets we curated for Mind2Web, Android in the Wild, and VisualWebArena, contain over 8k+, 4k+, and 900+ trajectories, respectively. We plan to review, clean, and release the complete trajectory datasets and the code to curate them. Also, we believe that the current subsets are sufficiently representative and reliable.
>
> > **W3**: Insufficient technical details about the implementation: Only cursory mention of using VNC and Docker No discussion of performance, latency, scalability, reliability considerations.
>
> Our implementation uses Docker for a lightweight environment compared to virtual machines used by OSWorld [1]. Key technical aspects include:
> - Performance & latency: The runtime mainly depends on the time for task reset, action generation & execution, auto-eval, and cleanup. AgentStudio is designed to mimic real-world computers operating in real time. The latency of reset, auto-eval, and cleanup depends on the Docker resource configuration, which can be improved by allocating additional resources to Docker. At present, the main bottleneck is the response time of LLM action generation, as well as the time required to execute LLM-generated code, rather than the latency in AgentStudio.
> - Scalability: Our architecture supports multiple instances running concurrently. The separation of the front end and back end ensures that LLM operations within Docker containers do not interfere with each other. Google API operations can also be isolated by setting up different accounts.
> - Reliability: Our Docker setup safeguards local files and systems from potential vulnerabilities. For sensitive operations, AgentStudio supports enabling human verification to enhance reliability and safety.

---

> > ### Author Response · Authors · 2024-11-18
> > **Response to Reviewer uDrN (2/2)**
> >
> > > **Q1**: Can AgentStudio enable large scale data collection, and agent evaluation? Will it support multi-threading that can be integrated to online RL training of agents?
> >
> > AgentStudio is designed with scalability in mind. Our tools facilitate extensive UI grounding data collection and video action annotation for scaling up benchmark and training data. With Docker, AgentStudio can run multiple environments in parallel for online RL agents, if needed.
> > However, the real-time nature and sparse rewards in our environment might present challenges for online RL methods, so it is not specifically optimized for online RL.
> >
> > > **Q2**: What are the key technical differences between AgentStudio and existing environments like VisualWebArena?
> >
> > We addressed this in our response to W1. Unlike VisualWebArena [2], which is browser-specific, our environment supports any software by its generic observation/action spaces, offering broader applicability. AgentStudio also introduces new research dimensions, e.g., selecting between API and GUI actions, decomposing agent abilities, which are not addressed by VisualWebArena.
> >
> > > **Q3**: Why are the evaluation datasets so small? Are there plans to scale them up?
> >
> > As addressed in our response to W2, the current dataset sizes reflect a balance between costs and comprehensiveness. We are committed to scaling our evaluation datasets by releasing the full trajectory datasets (totaling 13K+). This will help researchers expand IDMBench and CriticBench as needed.
> >
> > > **Q4**: In IDM-Multiple, accuracy is calculated based on the exact match of the action sequences. Is this too restrictive? Would there be multiple optimal action sequences? Even suboptimal, but successful sequences demonstrate agent capabilities.
> >
> > In addition to exact matches, we use the edit distance between predicted and ground truth action sequences as the metric (Table 9 of Appendix A). This metric evaluates the similarity between action sequences, providing a less restrictive evaluation than exact matches.
> >
> > Unlike task-completion datasets, where multiple action sequences can lead to the same goal state, in IDM-Multiple, each input image sequence corresponds to a single correct action sequence. This is because, given two observations, the action between them are deterministic within our provided action space. Examples are illustrated in Figures 16 and 17 of Appendix D.
> >
> > > **Q5**: The paper presents an interesting system but falls short in terms of technical novelty and scale of evaluation. The main contribution appears to be integration of existing components rather than fundamental technical advances. The small scale of the evaluation datasets also limits the impact. While the direction is promising, the current work would benefit from more technical depth and larger-scale evaluation.
> >
> > We believe that our responses to W1, W2, Q2, and Q3 adequately address the above concerns regarding novelty and dataset scale. We hope these clarifications lead to a favorable reassessment of our paper, and we are available to provide any further information.
> >
> > [1] Xie, Tianbao, et al. "OSWorld: Benchmarking multimodal agents for open-ended tasks in real computer environments." NeurIPS 2024 Datasets and Benchmarks Track.
> >
> > [2] Koh, Jing Yu, et al. "VisualWebArena: Evaluating multimodal agents on realistic visual web tasks." ACL 2024.

---

> ### Author Response · Authors · 2024-11-25
> **Follow-up on Response to Reviewer uDrN**
>
> We sincerely thank the reviewer for taking the time to evaluate our submission. As we approach the end of the discussion phase, we would greatly appreciate feedback on whether our responses have sufficiently addressed the concerns raised.
>
> - In our rebuttal, we clarified the technical contributions of our work, which have also been acknowledged in the reviews of other reviewers.
> - Regarding using a moderately-sized test set for IDMBench and CriticBench, this is a decision to ensure user-friendly and efficient benchmarks. Nevertheless, we will also provide a large-scale trajectory dataset to complement this. Importantly, our current experimental results of IDMBench and CriticBench remain robust, as recognized by other reviewers, including Reviewer w3vg ("providing a structured approach to evaluating and improving fundamental agent capabilities"), Reviewer v3Q2 ("The dataset curation process makes sense"; "The experiments are adequate"), and Reviewer iPnp ("advancing the field with nuanced benchmarks​").
> - Regarding implementation details, we have already included detailed information in our response and revised paper (see Appendix F.9). If there are any additional aspects that require clarification, we would be more than happy to provide further details.
>
> Thank you again for your review, and we hope our responses have addressed your concerns.

---

> > ### Comment · Reviewer_uDrN · 2024-11-28
> > **still have a number of questions**
> >
> > Thanks a lot for the rebuttal. I still have a number of questions.
> >
> > 1. "AgentStudio is designed with scalability in mind. Our tools facilitate extensive UI grounding data collection and video action annotation for scaling up benchmark and training data. With Docker, AgentStudio can run multiple environments in parallel for online RL agents, if needed. However, the real-time nature and sparse rewards in our environment might present challenges for online RL methods, so it is not specifically optimized for online RL."
> >
> > Have you done any scaling experiments, e.g. concurrently launching x number of machines and collecting y number of trajectories?
> >
> > 2. "Tools: We also provide tools for data collection, annotation, and benchmark creation, which helps expand benchmarks and scale up training data."
> >
> > The tools, e.g. annotation, from the UI, they seem to be very limited in features and usability. There are many good labeling tools, e.g. LabelStudio. Why not integrate them into your AgentStudo?
> >
> > 3 "We intentionally selected a representative subset of our trajectory datasets to ensure efficient and cost-effective evaluations for researchers. Since each trajectory contains multiple images, IDMBench and CriticBench are smaller than GroundUI to maintain a similar evaluation token cost. Notably, we have a larger trajectory dataset that could be utilized to scale up IDMBench and CriticBench. The original trajectory datasets we curated for Mind2Web, Android in the Wild, and VisualWebArena, contain over 8k+, 4k+, and 900+ trajectories, respectively. We plan to review, clean, and release the complete trajectory datasets and the code to curate them. Also, we believe that the current subsets are sufficiently representative and reliable."
> >
> > How did you ensure these datasets are representative? How many do you plan to release?
> >
> > 4. One additional question, the human success rate on Office is only 51%. Why so low? Are these professional annotators or just authors of the paper?

---

> > > ### Author Response · Authors · 2024-11-28
> > > **Response to Reviewer uDrN**
> > >
> > > Thank the reviewer for the follow-up questions. Our responses are as follows.
> > >
> > > > **Q1**: Have you done any scaling experiments, e.g. concurrently launching x number of machines and collecting y number of trajectories?
> > >
> > > We conducted a scaling experiment on a consumer-level Linux workstation (24-core CPU, 32 GiB memory) using naive actions (e.g., printing a string) to isolate environment performance from the overhead of executing complex actions (e.g., LLM-generated code). The experiment was performed with varying numbers of environments (from 1 to 16 Docker containers), and for each configuration, the results were the average of three independent runs.
> > >
> > > | # of envs | CPU usage (%) | Memory usage (MiB) | Action per second |
> > > | --------- | ------------- | ------------------ | ----------------- |
> > > | 1         | 2.01 ± 0.00   | 277.22 ± 0.03      | 87.04 ± 0.58      |
> > > | 4         | 3.84 ± 0.06   | 1103.57 ± 0.24     | 211.34 ± 2.38     |
> > > | 8         | 6.87 ± 0.25   | 2191.25 ± 1.04     | 280.99 ± 4.09     |
> > > | 16        | 10.87 ± 0.21  | 4356.37 ± 3.06     | 362.94 ± 3.55     |
> > >
> > > As the number of environments increases, the actions per second also grow, allowing more samples within the same wall time. Like many RL environments, scaling here is not linear due to resource contention and inter-process overhead. This can be further optimized using common techniques in RL environments like distributed rollouts with Ray [1] or cloud infrastructure [2]. Nevertheless, our primary focus is on LLM agents in real-time computer environments, where the interaction speed of the environment is not a bottleneck. As a result, we have chosen not to prioritize extreme optimization in this aspect.
> > >
> > > > **Q2**: The tools, e.g. annotation, from the UI, they seem to be very limited in features and usability. There are many good labeling tools, e.g. LabelStudio. Why not integrate them into your AgentStudo?
> > >
> > > Thank you for bringing this data annotation tool to our attention. We noticed that LabelStudio does not natively support the VNC interface, making it inefficient for simultaneous data collection and annotation in our setup. Unlike its two-step process of collecting and then labeling data, our tool allows data collection and annotation to happen at the same time. When designing our tools, we prioritized minimalism, out-of-the-box functionality, and avoiding third-party installations or configurations. Additionally, the video frame classification feature in LabelStudio—similar to our video action labeling—was released in version 1.14.0 after our paper submission. Nevertheless, we recognize LabelStudio as a highly useful tool, especially for detailed annotation of pre-collected, large-scale datasets. Moving forward, we plan to explore integrating compatible features to leverage its streamlined UI and more advanced functionalities.
> > >
> > > > **Q3**: How did you ensure these datasets are representative? How many do you plan to release?
> > >
> > > We constructed the IDM and CriticBench test sets based on the principle of maintaining class-balanced datasets [3, 4]. Specifically, for IDM, we ensured a balanced distribution of all action labels across the action space. Similarly, for CriticBench, we balanced successful and failed trajectories. We plan to further refine the full 13k trajectory dataset by filtering out erroneous or noisy samples and will release a clean version.
> > >
> > > > **Q4**: One additional question, the human success rate on Office is only 51%. Why so low? Are these professional annotators or just authors of the paper?
> > >
> > > The relatively low human success rate can be attributed to the difficulty of the Office tasks, which require more steps or specialized domain knowledge compared to those in OSWorld [5] (~70% success). Similar to OSWorld, our annotators are college students majoring in computer science, equipped with basic software usage skills but not professional expertise. While this setup ensures consistency across experiments and reflects the performance of typical human users, it does not represent the capabilities of human experts. Future work could involve recruiting professional annotators to establish an expert-level baseline.
> > >
> > > [1] Moritz, Philipp, et al. "Ray: A distributed framework for emerging AI applications." OSDI 2018.
> > >
> > > [2] Bonatti, Rogerio, et al. "Windows Agent Arena: Evaluating multi-modal OS agents at scale." arXiv preprint arXiv:2409.08264 (2024).
> > >
> > > [3] Torralba, Antonio, and Alexei A. Efros. "Unbiased look at dataset bias." CVPR 2011.
> > >
> > > [4] Recht, Benjamin, et al. "Do ImageNet classifiers generalize to ImageNet?." ICML 2019.
> > >
> > > [5] Xie, Tianbao, et al. "OSWorld: Benchmarking multimodal agents for open-ended tasks in real computer environments." NeurIPS 2024 Datasets and Benchmarks Track.

---

> ### Author Response · Authors · 2024-12-04
> **Follow-up on Addressed Issues**
>
> Dear Reviewer uDrN,
>
> Thank you for your valuable suggestions during the review and discussion phases. Your feedback has significantly improved the quality and clarity of our work.
>
> As you may have noticed, we have addressed your original concerns and provided detailed responses to your new questions regarding scalability, incorporating LabelStudio, sampling datasets, and human performance. We believe that these responses have effectively addressed your latest concerns.
>
> Although the discussion period has ended, we would like to kindly remind you that if you are willing to revisit your review and update your score, we would greatly appreciate it. Your reevaluation is important to us and would help ensure that the final assessment reflects our responses to your valuable feedback.
>
> Once again, thank you for the time you have dedicated to reviewing our work!

---

### Official Review · Reviewer_v3Q2 · 2024-11-03

**Soundness:** 3
**Presentation:** 3
**Contribution:** 3
**Rating:** 6
**Confidence:** 3

**Summary:**

In this paper, AgentStudio, a trinity of environments, tools, and benchmarks for building general virtual agents is proposed. The benchmark contains GroundUI that evaluates UI grounding capability, IDMBench that evaluates action labelling capability and CriticBench that evaluates success detection module. Overall experimental results show that the existing VLM is still far away from being able to fully solve these benchmarks. Compared w/ previous works, AgentStudio provides the better observation and action spaces, and therefore can help develop and evaluate agents in real-world settings.

**Strengths:**

1. The paper is well written, easy to follow.
2. The dataset curation process makes sense to me.
3. The experiments are adequate. Compared to previous works, AgentStudio has many advantages including interactivity, supporting data/tasks/tools, supporting language feedback, etc.
4. AgentStudio shows the short coming of existing models. For example, existing models can do pretty well on single API tasks, but very poorly on compositional tasks.
5. Also the benchmark shows that specialized models can do better than general model. For example, SeeClick does better than Gemini, Claude, etc for GroundUI.

**Weaknesses:**

1. Not sure # of tasks is enough. I would love to see the # of tasks can continue to grow.
2. Currently, the software seems to be randomly selected. One potential improvement is that maybe the author can get some statistics of most used software and include the top ones into the Benchmark. For example, I would imagine Photoshop could be an interesting case to add to the evaluation suite.

**Questions:**

1. Will AgentStudio be open sourced?
2. I did not quite get "For example, we can create a failure trajectory by extracting the first few steps of a successful one" How exactly is failure example being created?

---

> ### Author Response · Authors · 2024-11-18
> **Response to Reviewer v3Q2**
>
> Thank you for your valuable feedback. Below, we address each of your points in detail.
>
> > **W1**: Not sure # of tasks is enough. I would love to see the # of tasks can continue to grow.
>
> We believe that the current task count (205) is reasonable and comparable to related work. For example, Windows Agent Arena [1] has 154 tasks, and OSWorld [2] has 369 tasks. Our task count is a balanced tradeoff between evaluation efficiency, cost, and diversity.
>
> Moreover, AgentStudio is designed to be easily expanded in several ways:
> - **Template instantiation**: Our framework can increase the number of tasks by instantiating existing templates with different keywords. For example, WebArena [3] curated 241 templates, averaging 3.3 tasks per template, and VisualWebArena [4] collected 314 templates, averaging 2.9 tasks per template.
> - **Task composition**: Because AgentStudio has over one hundred auto-eval functions, our tasks can be expanded flexibly by combining existing instructions and auto-eval functions. In contrast, WebArena and VisualWebArena implemented around ten auto-eval functions.
> - **AgentStudio tools**: Our tools support new task creation and validation and are compatible with various evaluation methods, including functional auto-evaluation, human evaluation, and model evaluation.
>
> > **W2**: Currently, the software seems to be randomly selected. One potential improvement is that maybe the author can get some statistics of most used software and include the top ones into the Benchmark. For example, I would imagine Photoshop could be an interesting case to add to the evaluation suite.
>
> Our software selection was based on systematic criteria:
> - **Open-source**: We select open-source software to avoid copyright issues and encourage community contributions.
> - **Popularity**: We include widely used tools, such as Google Workspace, which has over 3 billion users globally [5], and VS Code and LibreOffice, which are among the most downloaded tools in their respective categories [6,7]. GIMP (GNU Image Manipulation Program) has been downloaded millions of times worldwide [8], making it one of the most popular open-source image editors.
> - **Diversity**: To represent a broad range of real-world use cases, we select software across different domains, including OS components, office suites, code editors, and image editors.
>
> While Photoshop is indeed popular, it is proprietary software. Although our environment can support such software, we prioritize using ones that do not require licenses for running benchmarks. Therefore, we choose GIMP, a widely-used open-source image editor with features comparable to Photoshop. Meanwhile, our benchmark tasks for GIMP are designed to be easily applicable to Photoshop as well, since most image-related tasks are evaluated by checking the output images.
>
> > **Q1**: Will AgentStudio be open sourced?
>
> Yes. AgentStudio has been open-sourced with comprehensive documentation, leaderboards, and accessible benchmark tasks and datasets. Due to the double-blind review policy, we are unable to provide URLs at this time, but all resources will be publicly available to the research community upon acceptance.
>
> > **Q2**: I did not quite get "For example, we can create a failure trajectory by extracting the first few steps of a successful one" How exactly is failure example being created?
>
> For example, we can generate a failure trajectory by selecting the initial five steps from a successful trajectory consisting of ten steps. This partial sequence leads to incomplete or incorrect outcomes, serving as a failure example.
>
> [1] Bonatti, Rogerio, et al. "Windows Agent Arena: Evaluating multi-modal os agents at scale." arXiv preprint arXiv:2409.08264 (2024).
>
> [2] Xie, Tianbao, et al. "OSWorld: Benchmarking multimodal agents for open-ended tasks in real computer environments." NeurIPS 2024 Datasets and Benchmarks Track.
>
> [3] Zhou, Shuyan, et al. "WebArena: A realistic web environment for building autonomous agents." ICLR 2024.
>
> [4] Koh, Jing Yu, et al. "VisualWebArena: Evaluating multimodal agents on realistic visual web tasks." ACL 2024.
>
> [5] [Google Workspace User Stats](https://explodingtopics.com/blog/google-workspace-stats).
>
> [6] [VS Code Counter](https://marketplace.visualstudio.com/items?itemName=uctakeoff.vscode-counter).
>
> [7] [LibreOffice - Wikipedia](https://en.wikipedia.org/wiki/LibreOffice).
>
> [8] [GNU Image Manipulation Program - Flathub](https://flathub.org/apps/org.gimp.GIMP).

---

> > ### Comment · Reviewer_v3Q2 · 2024-11-26
> >
> > Thanks for the reply. I will keep the rating unchanged.

---

### Official Review · Reviewer_w3vg · 2024-11-03

**Soundness:** 3
**Presentation:** 3
**Contribution:** 3
**Rating:** 6
**Confidence:** 3

**Summary:**

The paper introduces AgentStudio, a toolkit designed for building general virtual agents capable of handling multimodal observations and complex action spaces in dynamic, open-domain environments. It includes a set of environments, tools, and benchmarks that address the limitations of current domain-specific virtual agent evaluations. AgentStudio provides a lightweight, interactive environment with generic observation and action spaces, integrating tools for task creation, GUI element annotation, and video action labeling. The paper also presents three datasets—GroundUI, IDMBench, and CriticBench—that evaluate fundamental agent abilities such as GUI grounding, learning from videos, and success detection, aiming to advance the development of robust, general, and open-ended virtual agents.

**Strengths:**

-  This paper provides a lightweight, interactive environment with highly generic observation and action spaces, such as video observations and GUI/API actions, which expand the task space to a massively open domain and real-world tasks. AgentStudio comes with tools for creating and validating benchmark tasks, annotating GUI elements, and labeling actions in videos, which are essential for customizing and validating tasks in real-world settings.
-  The toolkit enables online interactions for learning through trial and error, providing language feedback on failure reasons, which is crucial for open-ended learning and self-improvement of LLM-based agents.
- The paper introduces three datasets—GroundUI, IDMBench, and CriticBench—that target UI grounding, action labeling from videos, and success detection, respectively, providing a structured approach to evaluating and improving fundamental agent capabilities.

**Weaknesses:**

-  Although the authors have made the code available in the supplementary materials, it would be beneficial to offer a more detailed guide to assist users in understanding and implementing the benchmark effectively.
- The paper's claims are somewhat overstated. While AgentStudio's tasks primarily focus on interactions within 2D graphical user interfaces (GUIs), the capabilities of a general virtual agent extend beyond these to include interactions with 3D virtual environments, such as those found in the metaverse. To align the paper's title with its scope or to enhance its benchmark, it would be essential to incorporate additional 3D world scenarios, like 3D video games, which would provide a more comprehensive assessment of a virtual agent's capabilities.

**Questions:**

How does AgentStudio handle user interactions, and what mechanisms are in place for agents to learn from and adapt to user feedback? Is it feasible to conduct a human evaluation of user experience when using the agent?

**Details Of Ethics Concerns:**

There is a genuine concern that such sophisticated tools, designed for interacting with digital devices and environments, could be exploited by malicious actors. Hackers might potentially misuse these capabilities to gain unauthorized access to personal or sensitive information, disrupt critical systems, or even conduct large-scale cyber attacks. This raises significant ethical concerns about privacy, security, and the potential for misuse, which must be proactively addressed.

---

> ### Author Response · Authors · 2024-11-18
> **Response to Reviewer w3vg**
>
> Thank you for your valuable feedback. Below, we address each of your points in detail.
>
> > **W1**: Although the authors have made the code available in the supplementary materials, it would be beneficial to offer a more detailed guide to assist users in understanding and implementing the benchmark effectively.
>
> We acknowledge the importance of comprehensive documentation. These resources will help users understand and use AgentStudio:
> - In our submission, we have included detailed data samples and prompts for our fine-grained evaluation and overall benchmark tasks in Appendices C-F. Appendix G provides more examples of AgentStudio tools.
> - In addition to the supplementary materials, we have created publicly available resources, including detailed READMEs and examples, to support the use of our benchmarks. (However, we cannot provide URLs here due to the double-blind review policy.)
>
> > **W2**: The paper's claims are somewhat overstated. While AgentStudio's tasks primarily focus on interactions within 2D graphical user interfaces (GUIs), the capabilities of a general virtual agent extend beyond these to include interactions with 3D virtual environments, such as those found in the metaverse. To align the paper's title with its scope or to enhance its benchmark, it would be essential to incorporate additional 3D world scenarios, like 3D video games, which would provide a more comprehensive assessment of a virtual agent's capabilities.
>
> We acknowledge the importance of 3D environments and agree that extending our work to such settings is an exciting direction for future research. Our paper clearly states the scope of our contributions, which are already more general than existing web and computer environments [1, 2, 3]. While our current benchmark tasks do not include 3D video games, the AgentStudio environment is designed with generic observation and action spaces, allowing straightforward extensions to other applications, including 3D video games. For example, we can evaluate agent performance in those games with video observations and GUI actions. Moreover, the fundamental abilities evaluated in our datasets—such as UI grounding, learning from videos, and success detection—are also crucial for virtual agents in 3D worlds.
>
> > **Q1**: How does AgentStudio handle user interactions, and what mechanisms are in place for agents to learn from and adapt to user feedback?
>
> In our implementation, users can interact with agents mainly in two ways: (i) by providing language feedback or (ii) by using a binary signal to indicate task completion. While the mechanisms by which agents learn from this feedback are beyond the scope of our work, we recognize several potential approaches, e.g., test-time self-correction via language feedback, learning a reward model from scalar reward, and training agents with reinforcement learning from environment feedback.
>
> > **Q2**: Is it feasible to conduct a human evaluation of user experience when using the agent?
>
> Yes, conducting a user experience evaluation is straightforward with AgentStudio tools. Our online benchmark visualization and human validation tool, as illustrated in Appendix G.4, is designed to facilitate the human evaluation of agents.
>
> > **Ethics Concerns**:  There is a genuine concern that such sophisticated tools, designed for interacting with digital devices and environments, could be exploited by malicious actors. Hackers might potentially misuse these capabilities to gain unauthorized access to personal or sensitive information, disrupt critical systems, or even conduct large-scale cyber attacks. This raises significant ethical concerns about privacy, security, and the potential for misuse, which must be proactively addressed.
>
> We fully acknowledge the ethical concerns of developing general virtual agents using AgentStudio. While our contributions focus on the infrastructure level rather than autonomous agents, we are committed to promoting the ethical use of our tools and have implemented several safeguards to prevent misuse. Specifically, we have designed our environment to be isolated from the user's physical machine, ensuring that agents cannot access private data. Additionally, we have incorporated optional human approval for risky actions, allowing users to monitor and validate agent behavior effectively. These measures mitigate potential privacy, security, and misuse risks.
>
> [1] Xie, Tianbao, et al. "OSWorld: Benchmarking multimodal agents for open-ended tasks in real computer environments." NeurIPS 2024 Datasets and Benchmarks Track.
>
> [2] Zhou, Shuyan, et al. "WebArena: A realistic web environment for building autonomous agents." ICLR 2024.
>
> [3] Koh, Jing Yu, et al. "VisualWebArena: Evaluating multimodal agents on realistic visual web tasks." ACL 2024.

---

> > ### Comment · Reviewer_w3vg · 2024-12-01
> >
> > Thanks for the reply. For the document, the publicly available resources can be shared in an anonymous URL link. For the ethics concern, I appreciate your efforts to provide measures in your benchmark to mitigate potential privacy, security, and misuse risks. However, are there any solutions to prevent the virtual agent be misused by others?

---

> > > ### Author Response · Authors · 2024-12-01
> > >
> > > Thank you for your valuable feedback.
> > >
> > > ---
> > >
> > > Regarding your concern about the document, we have included detailed README files in the supplementary materials, which provide comprehensive guidance for using and reproducing all benchmarks. The main README file in the root directory serves as an index to these documents. Please let us know if there are any specific aspects you would like us to elaborate on.
> > >
> > > ---
> > >
> > > Regarding your concern about preventing misuse of the virtual agent, we have implemented several security measures to mitigate potential risks. These measures align with best practices from the field, including those utilized in recent Claude 3.5 Sonnet's computer use [1]. Specifically, our safeguards include:
> > > 1. Containerization: The use of Docker containers to isolate the runtime environment and prevent direct system attacks or accidents.
> > > 2. Account Isolation: Operations requiring sensitive credentials are executed using isolated accounts to protect private information.
> > > 3. User Awareness and Consent: Users are explicitly informed of potential risks and must provide consent before engaging with our benchmarks.
> > > 4. Human Oversight: Actions that could result in real-world consequences require human confirmation to ensure ethical compliance.
> > >
> > > Additionally, most language model API providers and open models enforce strict terms of service (e.g., [2, 3, 4]) to prevent misuse, including prohibitions against violating laws or acceptable use policies. While we acknowledge the importance of more advanced approaches to ensure safe deployment (e.g., developing new classifiers to detect whether harm is occurring [1]), we believe that they are outside the scope of this study.
> > >
> > > ---
> > >
> > > We sincerely appreciate your suggestions and remain open to further clarifications. Please let us know if you have other concerns.
> > >
> > > [1] [Introducing computer use, a new Claude 3.5 Sonnet, and Claude 3.5 Haiku](https://www.anthropic.com/news/3-5-models-and-computer-use). Oct 2024.
> > >
> > > [2] [OpenAI Terms of Use](https://openai.com/policies/row-terms-of-use).
> > >
> > > [3] [Anthropic Consumer Terms of Service](https://www.anthropic.com/legal/consumer-terms).
> > >
> > > [4] [Mistral AI Legal Terms and Conditions](https://mistral.ai/terms).

---

### Meta-Review · Area_Chair_ppUu · 2024-12-17

**Metareview:**

The paper received acceptance ratings from all the reviewers (8,6,6,6). The paper provides a toolkit that integrates tools for creating benchmarks, annotating GUI elements, and labeling actions in videos. The toolkit is quite valuable for evaluating virtual agents’ capabilities across diverse software tasks. Hence, the AC concurs with the recommendation of the reviewers and recommends acceptance.

**Additional Comments On Reviewer Discussion:**

The reviewers initially had concerns such as overstated claims of the paper (Reviewer w3vg), insufficient task diversity (Reviewer v3Q2), small scale datasets (Reviewer uDrN), and simulated controlled environments (Reviewer iPnp). Overall, the rebuttal did not result in any score changes, except for reviewer uDrN, who indicated they would update their score. After extensive discussion, Reviewer uDrN still believed it is a borderline paper due to issues such as low success rate of the humans on the tasks and limitation of the features and usability of the tools. The AC checked the reviews and the responses. The responses are satisfactory in general. The AC agrees with the shortcomings mentioned by the reviewers, hence, a poster recommendation.

---

### Decision · Program_Chairs · 2025-01-22

Accept (Poster)